# Identification of PCPE-2 as the endogenous specific inhibitor of human BMP-1/tolloid-like proteinases

Sandrine Vadon-Le Goff [1,12], Agnès Tessier[1,2,3,12], Manon Napoli [1,12], Cindy Dieryckx[1], Julien Bauer[1], Mélissa Dussoyer[1], Priscillia Lagoutte[1], Célian Peyronnel[1], Lucie Essayan[1], Svenja Kleiser [2,3], Nicole Tueni [4,5,11], Emmanuel Bettler[1], Natacha Mariano[1], Elisabeth Errazuriz-Cerda[6], Carole Fruchart Gaillard[7], Florence Ruggiero [8], Christoph Becker-Pauly [9], Jean-Marc Allain [4,5], Leena Bruckner-Tuderman[2], Alexander Nyström [2,10] & Catherine Moali [1] ✉

BMP-1/tolloid-like proteinases (BTPs) are major players in tissue morphogenesis, growth and repair. They act by promoting the deposition of structural extracellular matrix proteins and by controlling the activity of matricellular proteins and TGF-β superfamily growth factors. They have also been implicated in several pathological conditions such as fibrosis, cancer, metabolic disorders and bone diseases. Despite this broad range of pathophysiological functions, the putative existence of a specific endogenous inhibitor capable of controlling their activities could never be confirmed. Here, we show that procollagen C-proteinase enhancer-2 (PCPE-2), a protein previously reported to bind fibrillar collagens and to promote their BTP-dependent maturation, is primarily a potent and specific inhibitor of BTPs which can counteract their proteolytic activities through direct binding. PCPE-2 therefore differs from the cognate PCPE-1 protein and extends the possibilities to fine-tune BTP activities, both in physiological conditions and in therapeutic settings.

BMP-1 (bone morphogenetic protein-1) and mTLL-1 (mammalian tolloid-like 1) are two essential extracellular metalloproteases for mammalian life. Mice lacking the *Bmp1* or *Tll1* gene die at mid-gestation or around birth with several defects affecting their skeleton, gut or heart[1,2]. Patients with mutations in their *BMP1* gene suffer from Osteogenesis Imperfecta[3]. These severe phenotypes are linked to the crucial contribution of the two enzymes to extracellular matrix assembly and growth factor activation[4]. Importantly, they proteolytically remove the C-terminal propeptide of fibrillar procollagens and, in some cases, their N-terminal propeptide to decrease the solubility of collagen monomers and trigger fibrillogenesis. They also activate lysyl oxidases LOX and LOXL to promote collagen and elastin cross-linking,

[1]University of Lyon, CNRS UMR5305, Tissue Biology and Therapeutic Engineering Laboratory (LBTI), 69367 Lyon, France. [2]Department of Dermatology, Medical Faculty, Medical Center - University of Freiburg, 79104 Freiburg, Germany. [3]University of Freiburg, Faculty of Biology, 79104 Freiburg, Germany. [4]Laboratoire de Mécanique des Solides, CNRS, Ecole Polytechnique, Institut Polytechnique de Paris, 91120 Palaiseau, France. [5]INRIA, 91120 Palaiseau, France. [6]University of Lyon, Centre d'Imagerie Quantitative Lyon-Est (CIQLE), SFR Santé-Lyon Est, 69373 Lyon, France. [7]Université Paris-Saclay, CEA, INRAE, Médicaments et Technologies pour la Santé (MTS), SIMoS, 91191 Gif-sur-Yvette, France. [8]ENS Lyon, CNRS UMR 5242, Institut de Génomique Fonctionnelle de Lyon (IGFL), 69007 Lyon, France. [9]University of Kiel, Biochemical Institute, Unit for Degradomics of the Protease Web, Kiel, Germany. [10]Freiburg Institute for Advanced Studies (FRIAS), University of Freiburg, Freiburg, Germany. [11]Present address: Institute of Applied Mechanics, Department of Mechanical Engineering, Friedrich-Alexander-Universität Erlangen-Nürnberg, 91058 Erlangen, Germany. [12]These authors contributed equally: Sandrine Vadon-Le Goff, Agnès Tessier, Manon Napoli. ✉e-mail: catherine.moali@ibcp.fr

process laminin 332 and collagen VII to allow basement membrane assembly and cleave dentin matrix protein-1 and dentin sialophosphoprotein to favor calcium fixation during bone and tooth mineralization. Other critical functions include the activation of BMP-2/4/11 growth factors through the cleavage of their antagonists or propeptides and the enhancement of TGF-β signalling through various cooperative mechanisms[4,5]. In mammals, the *BMP1* gene encodes two major splice variants named BMP-1 (or BMP-1-1) and mammalian tolloid (mTLD or BMP-1-3). Together with mTLL-2, which has been more specifically involved in the control of muscle growth through the cleavage of myostatin[6] but seems to have a more limited tissue distribution, BMP-1, mTLD and mTLL-1 form the BTP (BMP-1/tolloid-like proteinases) family that is included in the astacin-like subgroup of metzincins[7].

Strict control of BTP activities in time and space is also required for tissue homeostasis and regeneration in adults. For example, BTP activity is needed for bone remodelling[8] and BMP-1 dysregulation has been implicated in osteoporosis and osteoarthritis[9,10]. In soft tissues, skin wound healing is severely impaired in mice in the absence of *Bmp1* and *Tll1*[11]. Interestingly also, BTP expression is generally increased in fibrotic and scarring processes[12–14] while the inhibition of BTP activity tends to normalize tissue repair[15,16]. In addition, polymorphisms in *TLL1* were associated with the complications of some metabolic disorders (type 2 diabetes and non-alcoholic fatty liver disease)[17,18] and BMP-1 activity was suggested to play a role in lipid metabolism[19,20] and maternal diabetes[21]. Finally, high *BMP1* expression is a factor of poor prognosis in several cancer types[22,23] but the secretion of active BMP-1 by tumor cells in PDAC (pancreatic ductal adenocarcinoma) could be protective against tumor growth and metastasis[24].

Despite these important pathophysiological functions, the regulatory mechanisms controlling BTP activity have remained elusive. Some natural inhibitors have been described but these were either not specific to BTPs (α2-macroglobulin[25]), not conserved in mammals (sizzled[26,27]) or highly controversial with divergent effects observed in different laboratories (sFRP-2[26,28,29] and BMP-4[30,31]). Interestingly, this lack of specific inhibitors in mammals is associated with the existence of several enhancing proteins which, in most cases, seem to work in a substrate-specific manner to specifically promote the cleavage of one type of substrates (i.e. fibrillar collagens, LOX, myostatin and chordin)[4].

The most well-described BTP enhancer is PCPE-1 (procollagen C-proteinase enhancer 1) which efficiently stimulates the C-terminal maturation of fibrillar collagens by BTPs. In mice lacking PCPE-1 (*Pcolce*-null mice), aberrantly-shaped collagen fibrils were observed in bones and tendons and tissue mechanical properties were found to be altered[32]. Biochemical and structural studies have revealed that PCPE-1 exerts its procollagen C-proteinase (PCP) enhancing activity by directly binding the procollagen substrate in the C-terminal globular region (C-propeptide)[33]. This tight PCPE-1/procollagen interaction induces a local conformational change in the vicinity of the cleavage site, that facilitates the cleavage of the compact procollagen trimer by BTPs[34]. Another PCPE protein, called PCPE-2, shares 43% sequence identity with PCPE-1 and has the same domain structure consisting of two CUB (Complement, Uegf, BMP-1) domains followed by a linker and an NTR (netrin-like) domain[35]. It has been much less studied but, as it was also found to enhance the in vitro cleavage of procollagen II by BMP-1 and mTLL-1[36], the current view is that PCPE-2 behaves like PCPE-1 in terms of enhancing procollagen maturation.

Here, we analyse in more detail the role of PCPE-2 in collagen maturation and assembly, both in vivo and in vitro. We show that PCPE-2, in contrast to PCPE-1, has no significant effect on collagen organization and processing in *Pcolce2*-null mice. However, using in-depth biochemical analysis, we also demonstrate that it has the ability to form a tight complex with BMP-1 and to inhibit its proteolytic activity. This specific feature endows PCPE-2 with the possibility to fine-tune BTP activities and identifies PCPE-2 as the endogenous specific inhibitor of mammalian BTPs.

## Results

### *Pcolce2*-null mice do not show any major defects in collagen fibrils and PCPE-2 does not stimulate the procollagen I processing activity of BMP-1 in fibroblast medium

*Pcolce2*-null mice are viable and fertile and do not show any obvious phenotype in the absence of challenge[37–39]. In preliminary experiments, we checked that there was no compensation of the absence of PCPE-2 by PCPE-1 in these mice and that the expression of the BTPs involved in procollagen maturation was not affected by the lack of PCPE-2 (Supplementary Fig. 1). Then, we took advantage of this model to determine if PCPE-2 was important for the homeostasis of tissues rich in collagen I, as is the case for PCPE-1[32].

First, we analyzed the organization and morphology of collagen fibrils in the skin of wild-type (WT) and *Pcolce2*-null (KO) mice by transmission electron microscopy (TEM). In both genotypes, fibroblasts were surrounded by intertwined bundles of oriented collagen fibrils (Fig. 1a, Supplementary Fig. 2), which are typical of skin, and there was no sign of fibril fusion or irregular shape. We also measured the diameter of more than 3000 fibrils in TEM pictures of WT and KO skin and found that the median of the distribution was shifted by no more than 5 % and that the difference between the two genotypes was not significant when median diameters were compared for individual mice (Fig. 1b). Similar results were obtained in the heart, despite the fact that it is the major site of *Pcolce2* expression[35–37], and in tendons, where 100% of collagen fibrils were previously found to present irregular (scalloped) shapes in the absence of PCPE-1[32] (Supplementary Figs. 2, 3). Finally, we also performed uniaxial traction assays, as previously described for mouse skin[40], to determine if the absence of PCPE-2 had any impact on skin mechanical properties. The tangent modulus, heel region length, failure stretch and ultimate tensile stress were measured for WT and KO mice (Supplementary Fig. 4) but none of these parameters differed significantly between the two types of mice, indicating that PCPE-2 does not play a major role in the control of skin collagen organization and mechanical properties.

To get further insights into the effect of PCPE-2 on procollagen maturation, we extracted fibroblasts from mouse and human skins and compared the relative levels of mRNA present in these cells for PCPE-1 and PCPE-2. Interestingly, the mean *PCOLCE* gene expression was more than 40-fold higher than the mean *PCOLCE2* gene expression in fibroblasts from both species (Fig. 1c), confirming the previous result obtained at the protein level by Xu and colleagues who found that PCPE-1 was much more abundant than PCPE-2 in human fibroblast cell extracts[35]. We then analyzed procollagen I processing in the conditioned medium of WT and *Pcolce2*-null fibroblasts 8 h after serum starvation. In contrast to what has been previously described for *Pcolce*-null mouse embryonic fibroblasts (MEFs) and corneal keratocytes[32,41] but in agreement with the low abundance of PCPE-2 in dermal fibroblasts, there was no delay in the C-terminal maturation of procollagen I, as revealed by similar ratios of C-propeptide I to uncleaved precursors (procollagen I and pC-collagen I) in mouse fibroblast supernatants (Fig. 1d).

To get a better understanding of its activity, we also produced recombinant PCPE-2 fused to a 6 Histidine-tag at the N-terminus, in 293-T cells, and purified the protein by successive steps of heparin sepharose and Ni-NTA agarose chromatographies. The resulting protein was more than 95 % pure (Supplementary Fig. 5a), was not contaminated by endogenous PCPE-1, as confirmed by immunoblotting (Supplementary Fig. 5b), and displayed a circular dichroism spectrum similar to that of PCPE-1 with a negative peak around 213 nm[42] (Supplementary Fig. 6a). Recombinant PCPE-2 was then incubated with the supernatant of human adult fibroblasts in the presence of BMP-1 and its PCP enhancing activity on procollagen I was compared to that of

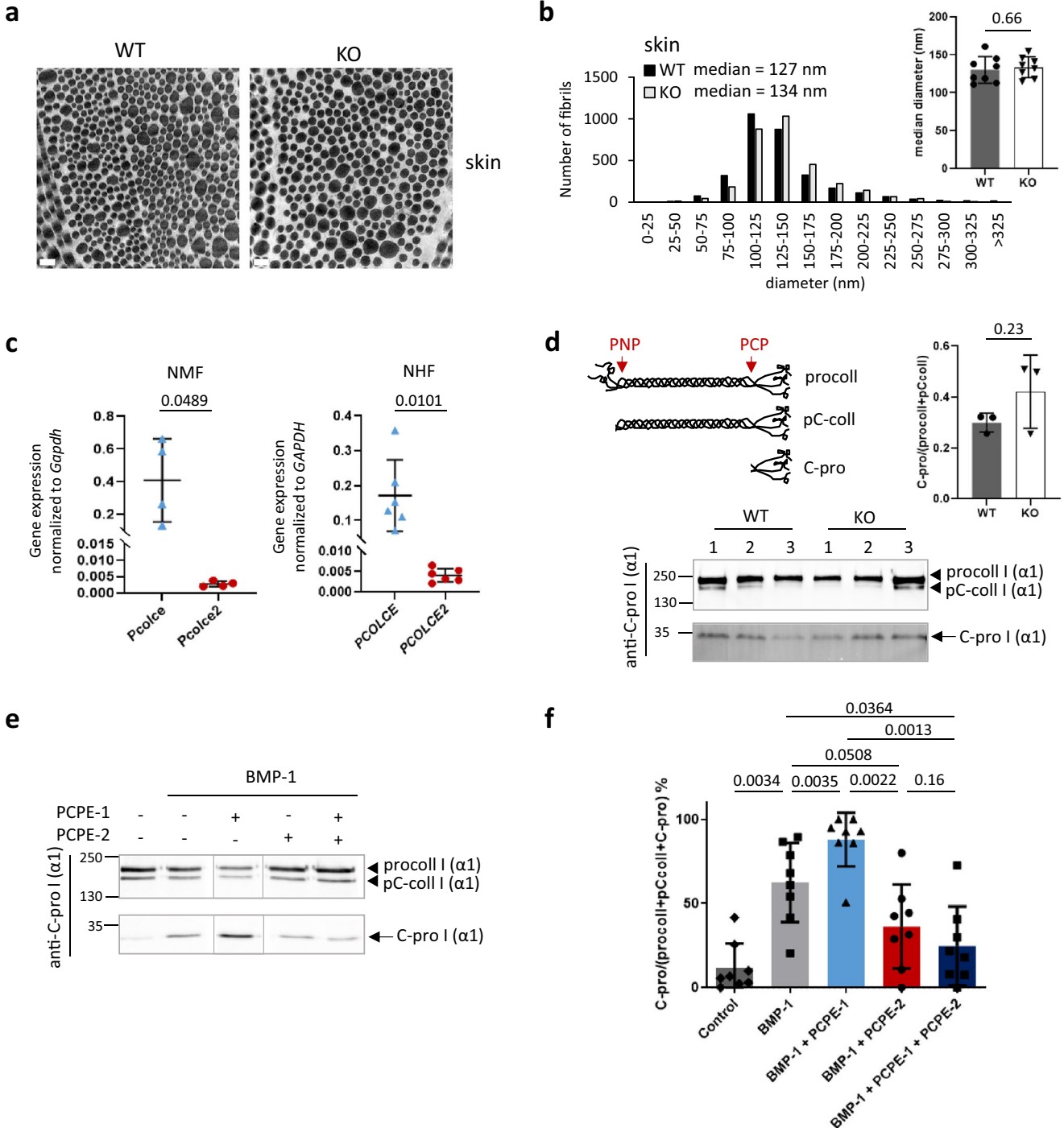

**Fig. 1 | The absence of endogenous PCPE-2 does not have a major impact on collagen fibrils in vivo but exogenous PCPE-2 regulates procollagen I maturation in fibroblast supernatants. a** TEM pictures of collagen fibrils in the skin of WT and *Pcolce2*-null (KO) mice (6 week-old). Representative images of *n* = 8 mice/genotype. Bar = 200 nm. **b** Distribution of skin collagen fibril diameters for WT and KO mice and median values of all measurements; 3092 fibrils from *n* = 8 mice/genotype were analyzed. The graph of the median diameters calculated for individual mice is also shown (means ± SD; *n* = 8 mice/genotype; unpaired two-sided *t*-test). **c** Comparison of the mRNA abundance for the genes encoding PCPE-1 and PCPE-2 by qRT-PCR in normal mouse and human dermal fibroblasts (NMF and NHF respectively). Graphs show means ± SD (*n* = 4 mice or 6 human donors; unpaired two-sided *t*-test with Welch's correction). **d** Immuno-detection of pro-collagen I, pC-collagen I and C-propeptide of collagen I (see schematic for a defi-nition; PNP procollagen N-proteinase; PCP procollagen C-proteinase) with the anti-

C-propeptide I (LF-41) antibody in the supernatant of WT and KO skin fibroblasts after 8 h of culture without serum. SDS-PAGE was run in reducing conditions and the same membrane was exposed for 1 min to detect the pro- and pC-collagens and for 3 min to detect the C-propeptide. Quantification of band intensities is shown for 3 mice/condition (means ± SD; unpaired two-sided *t*-test). **e** Detection of the same procollagen I processing products in the supernatant of human adult fibro-blasts after incubation for 2 h at 37 °C with buffer or 10 nM exogenous BMP-1. When indicated, 750 nM PCPE-1 and/or PCPE-2 were also added to the cleavage assays. **f** Quantification was with *n* = 8 independent experiments (conditions shown in the same order as in (**e**)). Means ± SD are shown and conditions were compared using one-way ANOVA with matched data (Geisser-Greenhouse correction) and Tukey's post-test. Uncropped immunoblots and source data for all graphs are available in Supplementary Fig. 16 or as a Source Data file.

recombinant PCPE-1 in the same conditions. While PCPE-1 addition led to a significant increase in C-propeptide release when compared to the condition with BMP-1 alone, PCPE-2 was unable to enhance BMP-1 PCP activity and even displayed a tendency towards inhibition (Fig. 1e, f). Even more surprising, PCPE-2 seemed to counteract the activity of PCPE-1 as the simultaneous addition of equimolar amounts of PCPE-1 and -2 failed to reproduce the enhancement obtained by PCPE-1 alone (Fig. 1e, f).

## PCPE-2 efficiently binds procollagens but does not behave like a canonical procollagen C-proteinase enhancer

To get further insight into the behavior of PCPE-2, we ran a series of in vitro experiments with three different mini-procollagens corresponding to truncated versions of fibrillar procollagens I, II and III, lacking the N-propeptides and made of short triple helices and intact C-telopeptides and C-propeptides[43]. These mini-procollagens were cleaved by BMP-1 and their cleavages were efficiently stimulated by PCPE-1 when the latter was present in equimolar amounts with the substrate (Fig. 2a). Interestingly, when tested in the same conditions, the enhancing effect of PCPE-2 was hardly detectable with mini-procollagens I and III but was more clearly evidenced with mini-procollagen II (Fig. 2a). The enhancement factor, calculated from band intensities, as described in Methods, remained however substantially lower in all cases than with PCPE-1 which allowed 100% of processing for all mini-procollagens in the conditions of the assay.

Since the enhancing activity of PCPE-1 is known to depend on its molar ratio to procollagen, we next tried to vary the concentrations of PCPE-1 and PCPE-2 in the assay. To achieve this, we used another model substrate derived from procollagen III, called CPIII-Long, and previously found to be well-suited for the precise quantification of BMP-1 C-proteinase activity[33]. Whereas the enhancing activity of PCPE-1 reached a plateau between 0.5 and 1 equivalent (PCPE:CPIII-Long trimer molar ratio) and remained stable at the maximum cleavage rate with 4 equivalents (enhancement factor > 3), PCPE-2 activity also reached a plateau between 0.5 and 1 equivalent, at the relatively high enhancement factor of 1.7-1.8, but then decreased to the basal level defined by BMP-1 alone at 4 equivalents (Fig. 2b). Using an even larger range of PCPE:CPIII-Long ratios (from 0:1 to 6:1), we actually found that the effect of PCPE-2 on BMP-1 procollagen-processing activity followed a bell-shaped curve (Fig. 2c), with activation at lower concentrations (ratios from 0.1:1 to 1:1) and inhibition at higher concentrations (ratios ≥ 3:1). In the conditions of this experiment, the maximum enhancement factor with PCPE-2 was 1.3-fold and the maximum inhibition reached 50% at the 6:1 ratio while PCPE-1 activity remained stable at the enhancement factor of 1.7 for all ratios superior to 0.3:1. The different behaviors of PCPE-1 and -2 were not due to different positions of the His tag in the two proteins since another PCPE-2 construct with a C-terminal 8His tag (like in PCPE-1) had the same effect as the N-terminally-tagged construct (Supplementary Fig. 6b). These results indicated that PCPE-2 did not behave like the canonical PCPE-1 enhancer and that the enhancing activity of PCPE-2 was hindered by another process that was potent enough to abrogate BMP-1 stimulation and could even lead to the inhibition of procollagen cleavage.

Since we know from previous studies that the interaction of the CUB domains of PCPE-1 with procollagens is the main driver of its enhancing activity[34,44], we then looked if PCPE-2 could also bind procollagens. First, we observed that the PCPE-1 residues that were previously shown to be involved in the interaction with procollagen III were perfectly conserved in PCPE-2 (Supplementary Fig. 7). In addition, when a model of the CUB domains of PCPE-2 was superimposed on the structure of the complex between the C-propeptide of collagen III and the CUB domains of PCPE-1[34] (Fig. 2d), the two PCPEs seemed to adopt a similar conformation with a rmsd (root-mean-square deviation) between the atomic positions of alpha carbons in CUB domains of 3.25 Å. To confirm the binding of PCPE-2 to procollagens, we analyzed

the interaction of recombinant PCPE-2 with immobilized mini-procollagen III by surface plasmon resonance (SPR). We observed that PCPE-2 also interacted efficiently with mini-procollagen III (Fig. 2e), leading to a dissociation constant of 11 nM (inset in Fig. 2e) to be compared to 2.2 nM for PCPE-1 (Supplementary Fig. 8a). This indicated that PCPE-1 led to a slightly more stable complex with mini-procollagen III than PCPE-2. Like for PCPE-1, the main site of interaction of PCPE-2 on mini-procollagen III was found to be the C-propeptide and interaction with this region alone led to a dissociation constant of 45 nM (Supplementary Fig. 8b). Furthermore, we could demonstrate that both proteins bound to overlapping sites on mini-procollagen III as there was no substantial increase in the recorded SPR signal when PCPE-1 was injected on a mini-procollagen III surface which was already saturated with PCPE-2 or vice-versa (Fig. 2f, Supplementary Fig. 8c).

In order to identify the domains of PCPE-2 involved in its interaction with procollagens, we also produced the CUB and NTR domains of PCPE-2 separately, using a PCPE-2 construct containing an internal HRV 3C-protease cleavage site (named PCPE-2 3C hereafter) (Supplementary Fig. 5c, d; Supplementary Fig. 6a). After purification, these domains were compared to full-length PCPE-2 and PCPE-2 3C in a CPIII-Long cleavage assay (Fig. 2b and Supplementary Fig. 9). Interestingly, while the NTR domain had no impact on substrate processing, PCPE-2 3C and the derived CUB1CUB2 domains could fully recapitulate the activity of full-length PCPE-2. No further change in CUB1CUB2 activity was observed when the NTR domain was present concomitantly (Supplementary Fig. 9). In agreement with the observation that the CUB domains of PCPE-2 played the most important role, these bound to mini-procollagen III with a $K_D$ around 30 nM (Supplementary Fig. 10a) while no interaction between the NTR domain and mini-procollagen III was found (Supplementary Fig. 10b). This suggests that the CUB domains are responsible for PCPE-2 activity, similarly to what was previously described for PCPE-1[33,44]. Also, we can conclude that the interaction of PCPE-2 with procollagens alone is unlikely to explain the complex activity profile described in Fig. 2c as the main function-related features of PCPE-1 are conserved in PCPE-2.

## PCPE-2 is a potent inhibitor of BMP-1

In addition to its PCP enhancing activity, PCPE-1 is also known to stimulate the cleavage of the N-terminal (TSPN) domain of the α1 chain of procollagen V by BMP-1[45], another activity related to collagen fibril assembly. To determine if PCPE-2 could play a similar role, we monitored the release of the TSPN domain from a truncated form of the procollagen V N-propeptide, called pNα1(V)[45], by SDS-PAGE. As expected, more TSPN was detected when PCPE-1 was present (at molar ratios to pNα1(V) between 0.1:1 and 2:1) than when BMP-1 alone was incubated with the substrate (Fig. 3a). In contrast, there was no visible TSPN product when PCPE-2 was added at all the tested molar ratios, suggesting that PCPE-2 strongly inhibited BMP-1 activity on pNα1(V).

Even more surprisingly, we observed that PCPE-2 also reduced the processing of several other known BMP-1 substrates on which PCPE-1 had no effect. This was the case for thrombospondin-1 (TSP-1), a recently described matricellular substrate involved in the control of cell adhesion and TGF-β activation[5]. Its cleavage products became hardly visible when PCPE-2 was added to the assay at a 1:1 molar ratio (Fig. 3b). Similarly, the processing of betaglycan, another BMP-1 substrate acting both as a TGF-β receptor (TGF-β RIII) at the cell surface and as a TGF-β antagonist when it is released in the extracellular environment[46], was reduced in the presence of PCPE-2 (Fig. 3c). PCPE-2 also inhibited the cleavage of chordin, a major antagonist of the BMP-2 and BMP-4 growth factors[47], and of endorepellin, an angiostatic fragment of perlecan[48] (Fig. 3d, e). Finally, the BMP-1-dependent cleavage of LDLR[19], a receptor that mediates the endocytosis of LDL (low-density lipoprotein) particles, was completely blocked by PCPE-2 (Fig. 3f). In summary, PCPE-2 inhibited the cleavages of all the BMP-1 non-

 

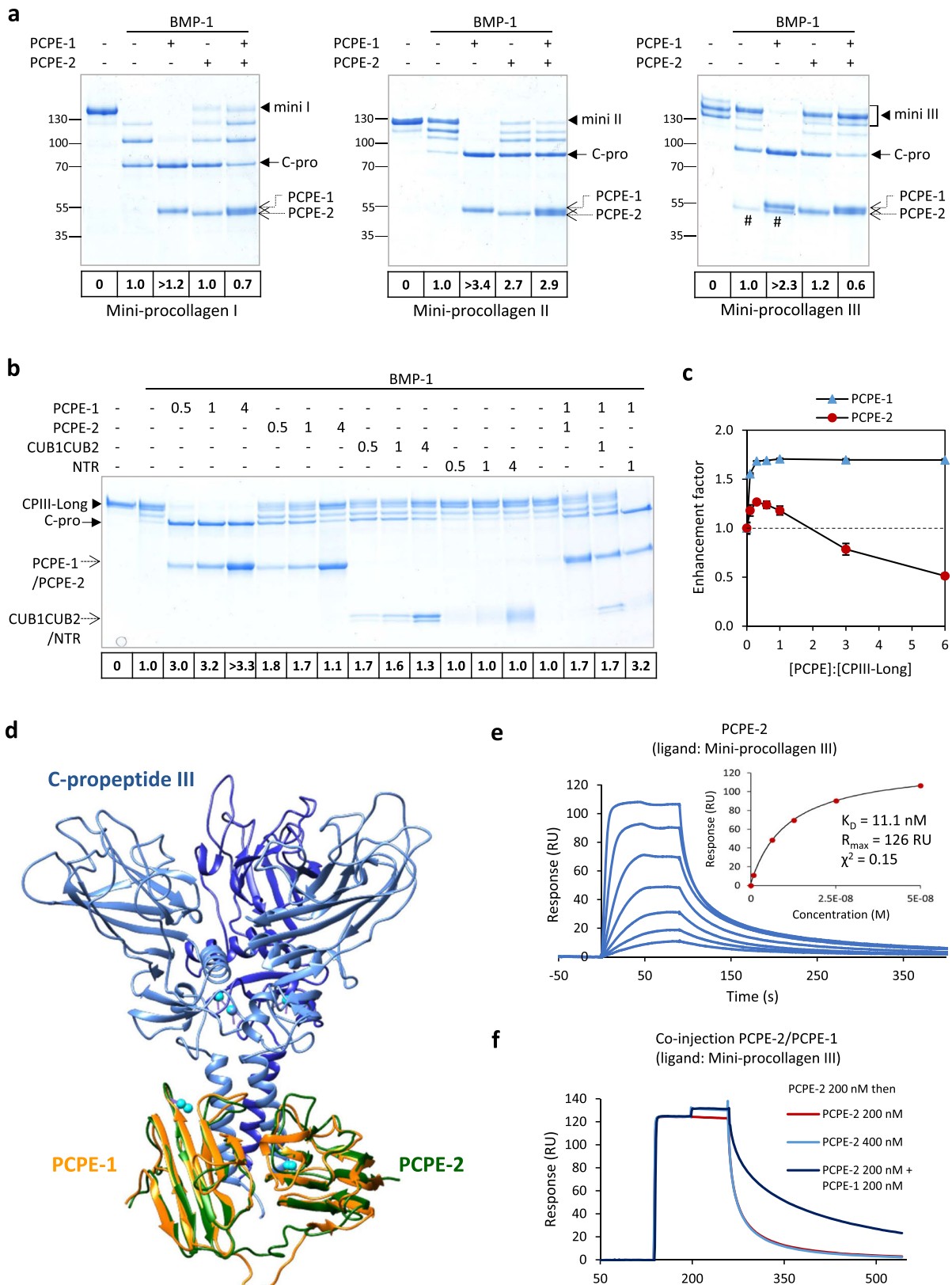

collagenous substrates that we were able to test, suggesting that it behaves as a broad-range inhibitor of BMP-1.

In order to check that this observed inhibitory activity of PCPE-2 was not an artifact of the protocol used to purify the recombinant protein, we looked at the processing of an endogenous substrate naturally expressed in 293-EBNA cells. Among the various possible

substrates that we tested, the C-terminal fragment of LDLR generated by BMP-1[19] could be readily detected in the cell lysate of 293-EBNA cells. The dependence on endogenous BMP-1 was confirmed by the comparison of three 293-cell lines stably expressing either PCPE-2, the corresponding empty vector or the *Xenopus* protein sizzled (not present in mammals) which is known to be a potent and specific

**Fig. 2 | The effect of PCPE-2 on procollagen C-terminal maturation is concentration-dependent and involves a direct interaction with procollagens.**
**a** Mini-procollagens I, II and III (430 nM) were incubated with 21 nM BMP-1 in the absence or presence of 430 nM PCPE-1 or/and PCPE-2 in a total volume of 30 μl for 1 h. Reaction products were analyzed by SDS-PAGE (4–20% polyacrylamide gel; non-reducing conditions) and Coomassie Blue staining. Enhancement factors estimated by densitometry are indicated below the gel. Gels are representative of $n = 3$ independent experiments. #Trimeric N-terminal cleavage product of mini-procollagen III (when visible). **b** Cleavage of CPIII-Long (360 nM) by BMP-1 (20 nM) for 1 h in the absence or presence of PCPE-1, PCPE-2, the CUB domains of PCPE-2, the NTR domain of PCPE-2 or combinations of PCPE-1 and PCPE-2 (or its domains). Molar ratios of PCPEs and domains to CPIII-Long are indicated above the gel and enhancement factors are reported below the gel. The gel is representative of $n = 3$ independent experiments. **c** Quantification of CPIII-Long (360 nM) processing by

BMP-1 (16.5 nM) in the presence of increasing amounts of PCPE-1 and PCPE-2 (0–2.2 μM). Analysis was by SDS-PAGE (reducing conditions) on a 15% acrylamide gel with Sypro Ruby staining and quantification of individual substrate bands (means ± SD; $n = 3$ independent experiments). **d** Superposition of a model of PCPE-2 CUB domains (green) on the X-ray structure (PDB code 6FZV) of the complex between the CUB domains of PCPE-1 (orange) and C-propeptide III (blue). Calcium ions are represented as cyan spheres. **e** PCPE-2 binding to immobilized mini-procollagen III (435 RU). Increasing concentrations of PCPE-2 were injected (0.78–50 nM, prepared as serial two-fold dilutions). The best fit of the binding response at equilibrium (steady-state conditions) is also shown (inset). Kinetic constants were outside instrument specifications and could not be determined. **f** Successive injections of PCPE-2 and PCPE-1 on immobilized mini-procollagen III (435 RU) compared to successive injections of PCPE-2 alone. Source data for all graphs are provided as a Source Data file.

exogenous inhibitor of human BTPs[26]. We found that LDLR C-terminal proteolytic product was clearly diminished in PCPE-2- and sizzled-transfected cells compared to cells transfected with the corresponding empty vector (Fig. 3g), confirming PCPE-2 inhibitory function in a cell-based assay.

Finally, the regulation of BMP-1 activity by PCPE-2 was assessed using the fluorogenic peptide Mca-YVADAPK(Dnp)-OH that can be cleaved by BMP-1 between the first alanine and the aspartate. Whereas the addition of PCPE-1 did not modify the cleavage rate of the peptide, PCPE-2 was also found to inhibit this cleavage at all tested concentrations (from 5 to 150 nM; Fig. 4a and Supplementary Fig. 6c). The apparent inhibition constant ($K_i^{app}$) value for PCPE-2 was calculated to be 2.8 ± 0.6 nM (Fig. 4b) which indicated that PCPE-2 was a tight-binding inhibitor of BMP-1[49]. However, it failed to decrease BMP-1 activity down to zero with around 25% residual activity at the highest PCPE-2 concentration (690 nM). This result suggested that PCPE-2 was not a competitive inhibitor directly binding to BMP-1 active site.

**The inhibitory activity of PCPE-2 is borne by its CUB domains and requires BMP-1 non-catalytic domains**
Our next goal was to decipher the mechanism of BMP-1 inhibition by PCPE-2. To do this, we used two constructs corresponding to CUB1 alone, prepared as a fusion with MBP (maltose-binding protein) that was subsequently removed by HRV 3C cleavage (Supplementary Fig. 5e, f), or CUB2NTR (Supplementary Fig. 5a) in addition to the separate CUB and NTR domains generated above. The CUB1 domain alone, the CUB2NTR region or the NTR domain alone had no effect on the cleavage of the fluorogenic peptide while the CUB1CUB2 sequence was as efficient at inhibiting BMP-1 as full-length PCPE-2 (Fig. 4a). This shows that the two CUB domains are necessary and sufficient for PCPE-2 activity. In the same conditions, neither PCPE-1 nor its CUB1CUB2 or NTR domains had any effect on the cleavage of the fluorogenic peptide.

In order to analyze the requirement for specific BMP-1 domains in the inhibitory mechanism by PCPE-2, we also generated successive deletion mutants of the protease either directly or through MBP fusions (Supplementary Fig. 11). All the BMP-1 constructs containing the catalytic domain could also cleave the fluorogenic peptide but only BMP-1[catCUB1CUB2] was affected by the presence of PCPE-2, to an extent similar to full-length BMP-1 (Fig. 4c). This shows that the minimal protease structure for inhibition must contain the first two CUB domains in addition to the catalytic domain. Moreover, the fact that the catalytic domain alone (BMP-1[cat]) was not inhibited by PCPE-2 also made unlikely the possibility that PCPE-2 acted as a competitive inhibitor binding into the protease active site and rather pointed to allosteric mechanisms affecting the catalytic domain.

In line with these results, PCPE-2, CUB1CUB2 and CUB2NTR but not NTR or CUB1 alone bound to immobilized BMP-1 by SPR (Fig. 4d–g, Supplementary Fig. 12a). These findings further suggest a major role of the CUB2 domain of PCPE-2 to drive complex formation. However,

cooperativity with CUB1 is required for activity (Fig. 4a) and helps to stabilize the complex, as illustrated by the faster dissociation of the BMP-1/CUB2NTR complex (Supplementary Fig. 12a) than of the BMP-1/CUB1CUB2 complex (Fig. 4g).

The dissociation constant for the interaction of PCPE-2 with BMP-1 was found to be 8.4 nM with the steady-state fit (Fig. 4f), in the same range as the $K_i^{app}$ measured above and as the dissociation constant for the interaction of PCPE-2 with procollagens (Fig. 2e). In the same conditions, PCPE-1 only gave a very weak signal (Fig. 4d), as previously described[50]. Interestingly, we also found that BMP-1[catCUB1CUB2] and BMP-1[CUB1CUB2] could bind PCPE-2 (Fig. 4e, Supplementary Fig. 12b) but that the interaction was lost for BMP-1[catCUB1] and BMP-1[cat], suggesting a major role of the CUB1CUB2 domains of BMP-1 in promoting the binding. Finally, a hydroxamate inhibitor blocking BMP-1 active site through zinc coordination did not prevent protease binding to PCPE-2 (Supplementary Fig. 12c), further supporting allosteric regulation of the catalytic activity of BMP-1 rather than direct competition with substrate binding in the active site.

All these findings establish that the difference in activity between the two PCPE proteins comes from the increased ability of PCPE-2 to form a stable complex with BMP-1, an interaction which further leads to BMP-1 inhibition.

**Interactions of PCPE-2 with procollagen and BMP-1 are mutually exclusive but binding sites are only partially overlapping**
To better understand the consequences of the dual binding capacity of PCPE-2, we ran a competition experiment which consisted of injecting increasing concentrations of BMP-1 on a C-propeptide III surface (an uncleavable substrate mimic) to which PCPE-2 was already bound (Fig. 5a). Interestingly, BMP-1 was able to disrupt the complex between PCPE-2 and C-propeptide III in a concentration-dependent manner, leading to 77% inhibition when both BMP-1 and PCPE-2 were injected at the same concentration (200 nM). Here again, PCPE-2 behaved differently from PCPE-1 as the simultaneous injection of PCPE-1 and BMP-1 over immobilized C-propeptide III did not lead to a decreased signal but rather to an increased signal (Fig. 5b), revealing that BMP-1 and PCPE-1 could bind together to C-propeptide III. In a similar experiment, we co-injected mini-procollagens I, II or III with PCPE-2 over immobilized BMP-1 and found that all mini-procollagens hindered PCPE-2 interaction with BMP-1 (Supplementary Fig. 13a). This showed that PCPE-2 could not bind simultaneously to the procollagens and to the protease and this seemed to apply to all major fibrillar procollagens. Moreover, the ability to bind substrates appeared specific to the process of procollagen maturation as PCPE-2 was not found to bind betaglycan or the ectodomain of LDLR, two other physiological BTP substrates (Supplementary Fig. 13b). These important results demonstrate that the interactions of PCPE-2 with the procollagen substrate and the BMP-1 protease are mutually exclusive and that a stable ternary complex involving BMP-1, PCPE-2 and a procollagen substrate is unlikely to occur. Also, this dual binding capacity does not apply to other

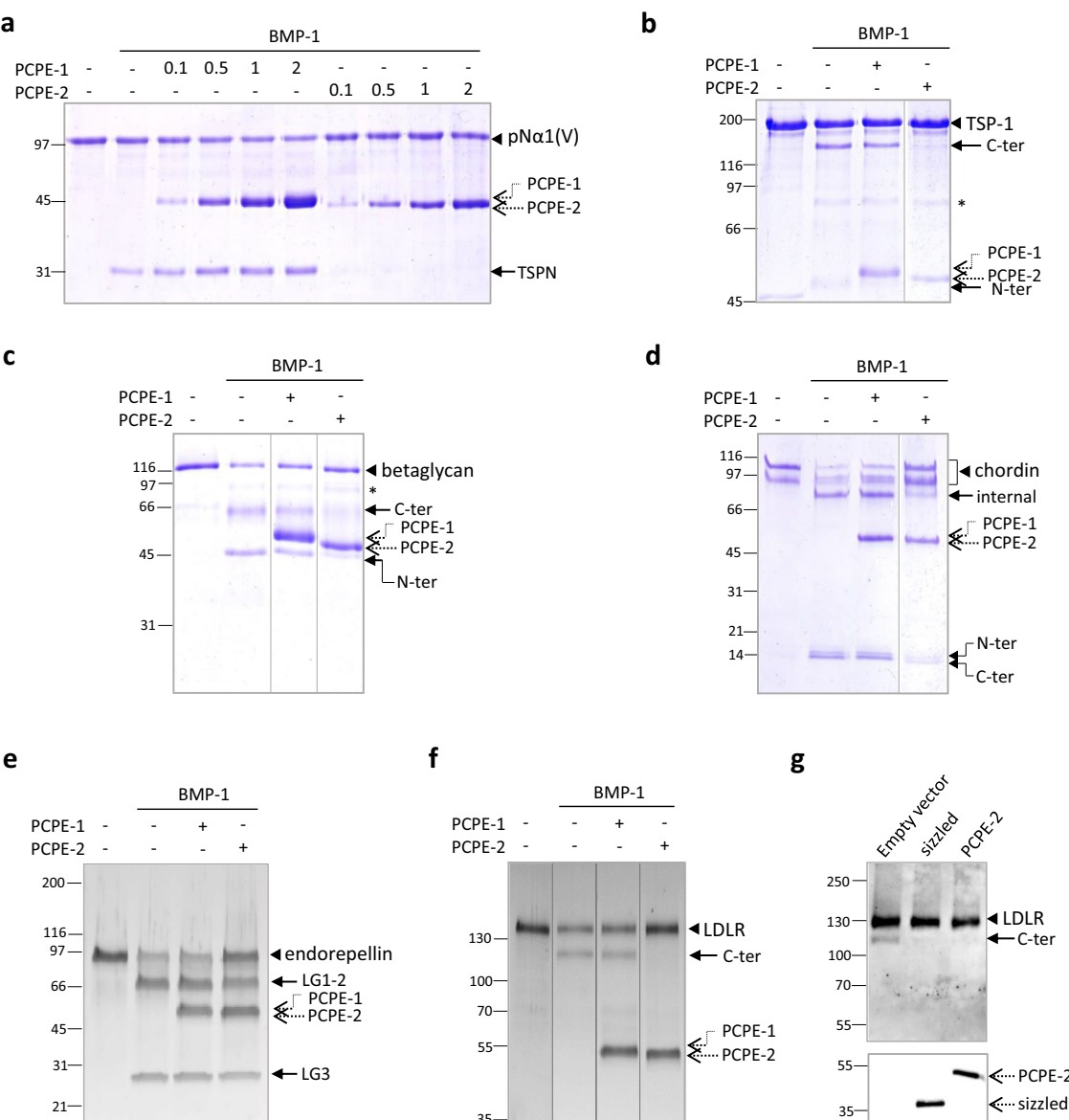

**Fig. 3 | PCPE-2 inhibits the cleavage of BMP-1 substrates. a–e** Effect of PCPE-1 and -2 on various BMP-1 substrates at 1:1 PCPE:substrate molar ratios (unless otherwise indicated) at 37 °C. **a** The pNα1(V) protein (800 nM) is the monomeric N-terminal region of the α1 chain of collagen V and it was incubated with 20 nM BMP-1 for 2 h to monitor the release of the N-terminal TSPN domain (PCPE to pNα1(V) molar ratios as indicated above the gel). **b** TSP-1 (200 nM) was incubated with 70 nM BMP-1 for 4 h to generate one N-terminal fragment (45 kDa) and one C-terminal fragment (120 kDa). **c** The ectodomain of betaglycan (780 nM) is cleaved by BMP-1 (50 nM) in two main products (N-ter, C-ter) in the conditions of the assay (4 h). **d** Chordin (present in two forms in the starting material, resulting from alternative splicing according to the manufacturer; 370 nM) yielded 3 products (referred to as N-ter, internal and C-ter) when incubated with 20 nM BMP-1 for 4 h. **e** Endorepellin (265 nM) corresponds to the C-terminal part of perlecan and was cleaved by BMP-1

(18 nM) in two fragments (LG1-2 and LG3) after 1 h. **f** The ectodomain of LDLR (390 nM) was converted to a shorter fragment (C-ter) in the presence of 1.5 nM BMP-1 for 1 h. In (**a**–**f**), detection was by SDS-PAGE (reducing conditions) with Coomassie Blue staining except for endorepellin and LDLR which were detected with Sypro Ruby staining. (*) indicates BMP-1 position when visible. The gels are representative of n = 4 independent experiments for (**a**), n = 2 for (**b**, **d**, **e**) and n = 3 for (**c**, **f**). **g** Immuno-detection of endogenous LDLR in 293-EBNA cell lysates, after stable transfection with an empty vector or the same vector containing sizzled sequence or PCPE-2 sequence. Expression of sizzled and PCPE-2 was also evidenced by immunoblotting with anti-His antibody in the supernatant of transfected cells. The immunoblots are representative of n = 2 independent experiments. Uncropped gels and blots can be seen in Supplementary Fig. 16 or as a Source Data file.

BMP-1 activities where PCPE-2 is only targeted to the protease and does not bind the substrate.

Since above results suggested significant overlap between binding sites for BMP-1 and procollagens on PCPE-2, we next tried to determine if the same residues were involved in these two types of interactions. Based on the conservation of the important residues for procollagen binding between PCPE-1 and -2 (Supplementary Fig. 7), we first mutated a phenylalanine known to play a key role in the PCPE-1/procollagen III interaction (F87 in PCPE-2)[34,51] in a MBP-CUB1CUB2 DNA template that

we generated for this purpose (Supplementary Fig. 5e, f). In an attempt to identify potential residues involved in BMP-1 binding, we also looked for charged residues that were conserved in PCPE-2 proteins from various mammalian species but specific to the CUB domains of PCPE-2 (Supplementary Fig. 14a, b). This pointed to 4 residues (K46 in CUB1 and R150, E197 and R220 in CUB2) which were mutated in two groups based on their relative positions in the CUB1CUB2 homology model (Supplementary Fig. 14c; R150A on one side and K46A/E197A/R220A on the other side). We found that F87 was actually critical for procollagen

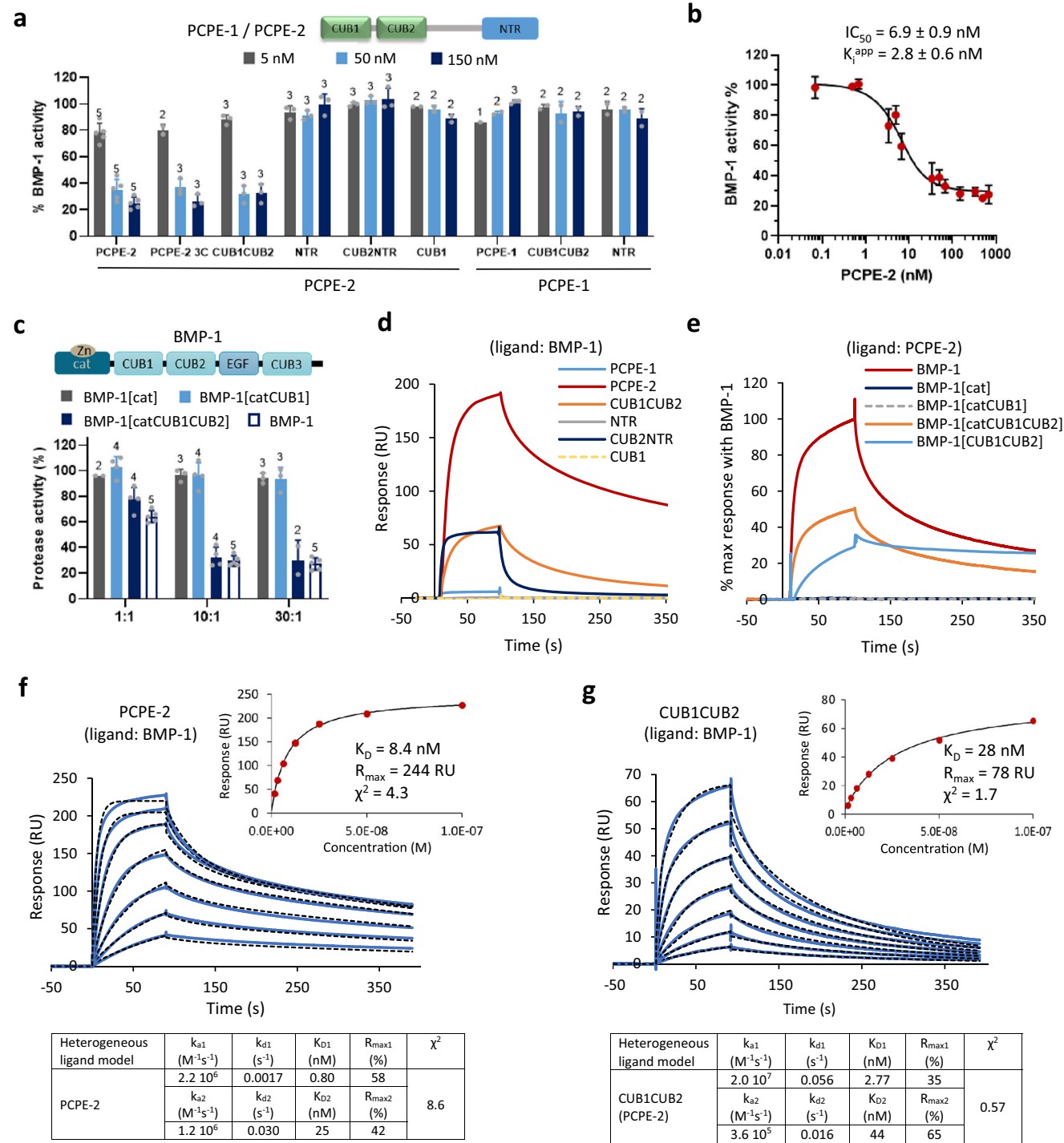

**Fig. 4 | The CUB domains of PCPE-2 are responsible for its inhibitory activity and the non-catalytic domains of BMP-1 are required for inhibition.**
**a** Quantification of BMP-1 activity on the fluorogenic peptide Mca-YVADAPK(Dnp)-OH in the presence of increasing concentrations of PCPE-2, PCPE-2 3C and various deletion constructs of PCPE-2 or PCPE-1. BMP-1 concentration in the experiment was 7 nM. Means ± SD (number of independent experiments run in duplicate indicated above each bar). **b** The $IC_{50}$ of PCPE-2 was measured in the same conditions with PCPE-2 concentrations between 0.07 and 690 nM and the corresponding apparent inhibition constant ($K_i^{app}$) was calculated using the Morrison equation. Means ± SD of $n = 3$ independent experiments run in duplicate. **c** Effect of PCPE-2 (at the indicated molar ratios to protease) on the cleavage of the

fluorogenic peptide by deletion mutants of BMP-1 (7 nM except for BMP-1[cat]: 8 nM). Means ± SD (number of independent experiments run in duplicate indicated above each bar). **d** Comparison of the binding of 50 nM of PCPE-1, PCPE-2 and its domains (CUB1CUB2, CUB2NTR, NTR and CUB1) on immobilized BMP-1 (974 RU). **e** Comparison of the binding of 50 nM BMP-1 and its deletion mutants on immobilized PCPE-2 (646 RU or 418 RU) after normalization with the maximum response obtained with BMP-1. **f** Fits of sensorgrams obtained with the kinetic (black dotted lines; heterogeneous ligand) or steady-state (inset) model when increasing concentrations of PCPE-2 (1.56-100 nM, prepared as serial two-fold dilutions) were injected over immobilized BMP-1 (974 RU). **g** Same as in (**f**) for the CUB1CUB2 domains of PCPE-2. Source data for all graphs are provided as a Source Data file.

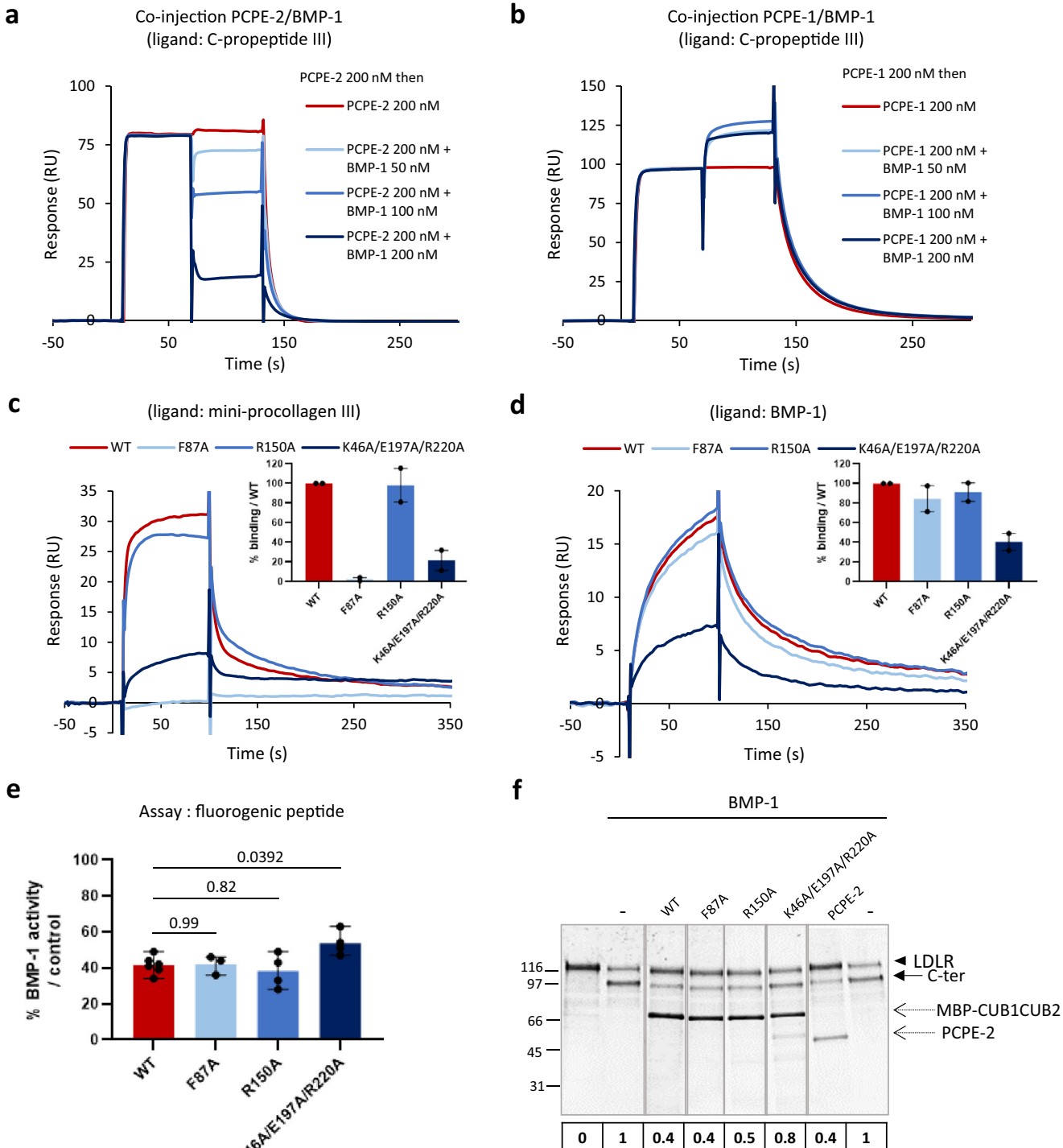

**Fig. 5 | Interactions of PCPE-2 with procollagen and BMP-1 are mutually exclusive but the binding sites are only partially overlapping. a** Competition experiments consisting of an injection of 200 nM PCPE-2 for 60 s followed by an injection of PCPE-2 (200 nM) alone or in combination with 50, 100 or 200 nM BMP-1 for another 60 s. The immobilized protein was C-propeptide III (351 RU). **b** Same as in (**a**) with PCPE-1 instead of PCPE-2. **c** Binding of WT and mutant CUB1CUB2* domains of PCPE-2 (50 nM, obtained from MBP-CUB1CUB2 fusion constructs) on immobilized mini-procollagen III (437 RU) as determined by SPR. Bar graph shows means ± range of $n = 2$ independent experiments performed in duplicates on two different surfaces with 437 and 651 RU of immobilized mini-procollagen III. **d** Binding of WT and mutant MBP-CUB1CUB2 domains of PCPE-2 (50 nM) on immobilized BMP-1 (1302 RU) as determined by SPR. Bar graph shows means ±

range of $n = 2$ independent experiments performed in duplicates on two different surfaces with 1302 and 1492 RU of immobilized BMP-1. **e** Quantification of BMP-1 activity on the fluorogenic peptide in the presence of WT and mutant MBP-CUB1CUB2 domains of PCPE-2 (% of BMP-1 activity relative to control obtained in the absence of MBP-CUB1CUB2 domains); means ± SD of $n = 4$ independent experiments performed in duplicates except for WT ($n = 6$) and F87A ($n = 3$). Mutants were compared to WT using one-way ANOVA and Dunnett's post-test. **f** Effect of WT and mutant MBP-CUB1CUB2 domains of PCPE-2 (388 nM) on the cleavage of LDLR ectodomain (388 nM) by BMP-1 (17 nM). Activities relative to the BMP-1 alone condition are indicated below the gel. The gel is representative of $n = 2$ independent experiments. Uncropped gel and source data for all graphs are available in Supplementary Fig. 16 or as a Source Data file.

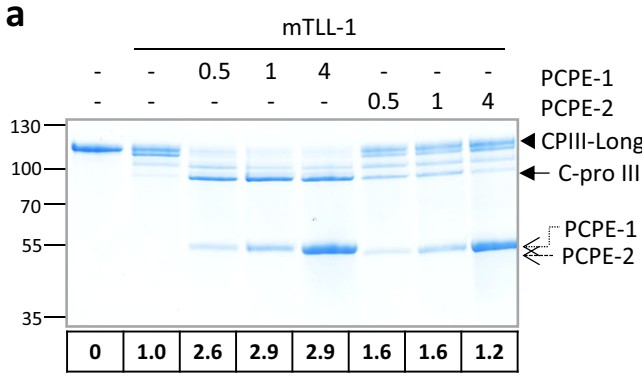

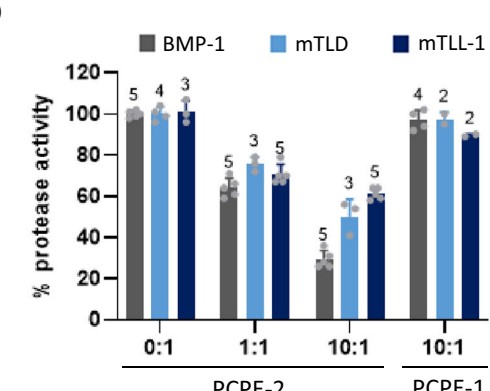

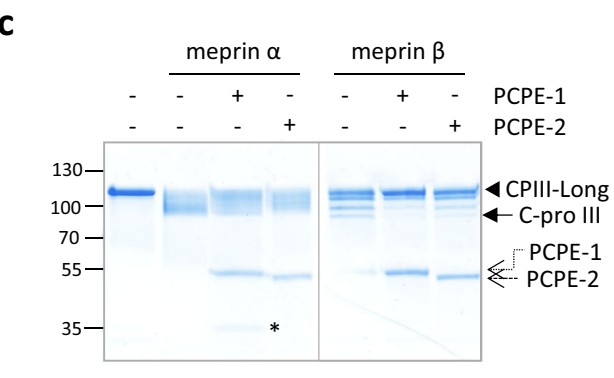

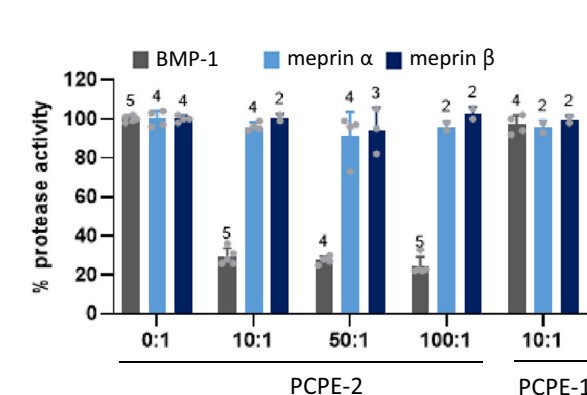

**Fig. 6 | Effect of PCPE-2 on other human astacins. a** SDS-PAGE analysis in non-reducing conditions of the cleavage of CPIII-Long (320 nM) by mTLL-1 (17 nM) in the absence or presence of PCPE-1 or PCPE-2 for 4 h. Molar ratios of PCPEs to CPIII-Long are indicated above the gel and the enhancement factors are indicated below the gel. The gel is representative of $n = 3$ independent experiments. **b** Comparison of BMP-1 (10 nM), mTLD (35 nM) and mTLL-1 (50 nM) activities on the fluorogenic peptide Mca-YVADAPK(Dnp)-OH, in the absence or presence of PCPE-1 or PCPE-2 (at the indicated molar ratios to protease). Means ± SD (number of independent experiments run in duplicate indicated above each bar). **c** SDS-PAGE analysis in non-reducing conditions of the cleavage of CPIII-Long (470 nM) by meprin α

(3 nM) or meprin β (1.5 nM), in the absence or presence of equimolar PCPE-1 or PCPE-2 concentrations (470 nM), for 15 min. The gel is representative of $n = 3$ independent experiments. (*) indicates the N-terminal cleavage product of PCPE-1 by meprins[52] when visible. **d** Comparison of BMP-1 (10 nM), meprin α (1 nM) or meprin β (0.5 nM) activities on the fluorogenic peptide Mca-YVADAPK(Dnp)-OH, in the absence or presence of PCPE-1 or PCPE-2. Means ± SD (number of independent experiments run in duplicate indicated above each bar). Uncropped gels and source data for all graphs are available in Supplementary Fig. 16 or as a Source Data file.

binding (Fig. 5c) but had little effect on the interaction with the protease and hence on the inhibitory activity of the CUB domains of PCPE-2, either with a peptide substrate or with a physiological substrate (LDLR) (Fig. 5d–f). If the R150A mutation did not affect procollagen or protease binding, the triple K46A/E197A/R220A mutant led to significant effects on both interactions and this translated into a reduced inhibitory potency of the CUB domains in the peptide and LDLR cleavage assays. In summary, the overlap between binding sites is only partial but sufficient to prevent the simultaneous interactions of the protease and procollagen partners with PCPE-2.

### PCPE-2 specifically inhibits BTPs but also interferes with the PCP activity of meprins through its interaction with procollagens

We next wanted to know if PCPE-2 was specific to BMP-1. We first tested its effect on mTLD and mTLL-1 in a CPIII-Long cleavage assay (Fig. 6a; Supplementary Fig. 15a). Interestingly, the enhancement of the PCP activity of mTLD or mTLL-1 by PCPE-2 followed the same trends as for BMP-1 with maximum enhancement factors reached between 0.5:1 and 1:1 molar ratios which then dropped at higher PCPE-2 concentrations. In addition, PCPE-2 could efficiently inhibit the two enzymes in the fluorogenic peptide assay (Fig. 6b), albeit to a slightly lower extent than with BMP-1. This indicated that PCPE-2 could also affect both the collagenous and non-collagenous

substrates of the tolloid enzymes, in agreement with what was observed above for BMP-1.

Meprin α and meprin β are two other members of the astacin-like subgroup of metalloproteinases that were tested next. They were differentially impacted in the fluorogenic peptide and CPIII-Long assays with no effect of PCPE-2 on the peptide cleavage but a slight inhibition of the maturation of the procollagen-derived protein by meprins (Fig. 6c, d). The latter result was in agreement with our previous finding that PCPE-1 could inhibit the PCP activity of meprins by blocking their access to the cleavage site through its tight interaction with procollagens[52]. A similar mechanism was probably also at play with PCPE-2 but to further confirm that PCPE-2 does not inhibit other meprin activities, another physiological substrate of these proteases (CD99[53]) was tested. As expected, CD99 cleavage was not affected by PCPE-2 (Supplementary Fig. 15b) and this strongly suggested that PCPE-2 inhibition specifically targets BTPs. Consequently, PCPE-2 appears as a specific inhibitor of BTPs in mammals.

## Discussion

Most protease families have their own specific inhibitors which protect cellular and extracellular proteins from the deleterious effects of a prolonged or excessive proteolytic activity. The well-described inhibitors of matrix metalloproteinases are the TIMPs (tissue inhibitors of

**a-** Non-collagenous BTP substrates

substrate · PCPE-1 · protease · **no effect**

substrate · PCPE-2 · protease · **inhibition**

**b-** Fibrillar collagens I-III

procollagen · PCPE-1 · protease · **enhancement**

procollagen · PCPE-2 · protease · **enhancement / no effect / inhibition**

**Fig. 7 | Overview of BTP regulation by PCPE-1 and PCPE-2. a** PCPE-1 has no effect on the BTP-dependent cleavage of non-collagenous substrates while PCPE-2 acts as a direct and potent inhibitor of these cleavages. Our results indicate that the CUB domains of PCPE-2 interact with BMP-1 auxiliary domains (CUB1 and CUB2) to allosterically inhibit the activity of the protease domain. **b** In the case of fibrillar procollagens, PCPE-1 enhances the C-terminal maturation of procollagens I-III through the binding of CUB1CUB2 to C-propeptides. PCPE-2 binds procollagens and BMP-1 with similar affinities but cannot interact with both partners simultaneously, thereby precluding the efficient processing of procollagens. The net effect (enhancement, no effect, inhibition) therefore depends on the relative concentrations of the three partners (procollagen, protease, PCPE-2). PCPE-1 and -2 are represented in their 2-domain (CUB1CUB2) configuration while BTPs are shown in the BMP-1 configuration (monomers with CUB1-CUB2-EGF-CUB3 auxiliary domains). Note that the incompletely-characterized case of fibrillar collagens V and XI is not represented.

metalloproteinases) but they are not active on BTPs[54]. Meprins are described to be regulated by fetuin-B[55] but again this protein is not effective against BTPs. Here, by demonstrating that PCPE-2 is a potent and specific inhibitor of BTPs targeting all their proteolytic activities, we put an end to the exceptional status of BTPs which seemed to rely on enhancing proteins more than on inhibitors to control their activity. This situation is actually more coherent with the fact that BTPs are activated during their transit in the trans-Golgi network and that they are supposedly fully active when they are secreted in the extracellular environment, with no other means than inhibition or degradation to abolish their activity.

We have shown in this study that the broad inhibition of BMP-1 activities by PCPE-2 is mediated by a direct and strong interaction between the regulatory protein and the protease, a mechanism that is not shared by PCPE-1. Rather surprisingly, PCPE-2 has a netrin-like (NTR) domain in common with the TIMPs but its inhibitory activity is not borne by this domain which is fully dispensable for inhibition. In contrast, the CUB domains of PCPE-2 can nicely recapitulate BMP-1 binding and inhibition, suggesting that these domains alone are necessary and sufficient. On the protease side, the catalytic domain of BMP-1 alone cannot interact with PCPE-2 and several results, including the absence of competition with a small-molecule inhibitor, suggest that PCPE-2 does not bind into the active site of the enzyme like most protease inhibitors. Actually, we found that the interaction between the two proteins is driven by the non-catalytic domains of BMP-1, especially CUB1 and CUB2. Whether this interaction alone is sufficient to block substrate processing or whether the interaction with non-catalytic domains helps to recruit the catalytic domain to form a

bivalent complex remains to be established. In both cases, allosteric mechanisms seem to play a major role and point to an original mode of action for PCPE-2.

A summary of the finely-tuned regulation of BTP activity permitted by the two PCPEs is presented in Fig. 7. The entire range of possible outcomes, from no effect to enhancement and inhibition, is potentially available but the final effect can vary with the nature of the substrate. In the case of non-collagenous substrates (Fig. 7a), the situation is rather straightforward with PCPE-1 having no effect and PCPE-2 behaving as a potent inhibitor of their BTP-dependent cleavage. As a result, only inhibition of proteolytic activity (by PCPE-2) can be observed. In contrast, for fibrillar procollagens, our results suggest a much more complex situation (Fig. 7b). If previous findings regarding the efficient enhancement of BTP activity by PCPE-1 were fully confirmed in this study, it is not the case for PCPE-2. Even if enhancement of procollagen maturation can be observed in vitro in a specific range of PCPE-2 concentrations, the level of stimulation achieved by this protein is generally much lower than by PCPE-1. This is especially true for procollagens I and III but less for procollagen II, for which the enhancement is more easily observed. Therefore, the use of the procollagen II substrate for the early characterization of PCPE-2[36] may have led to the partially flawed conclusion that, at least in vitro, PCPE-2 works exactly like PCPE-1. Indeed, we have observed that, in some situations, PCPE-2 (but not PCPE-1) can also inhibit the PCP activity of BMP-1 or have no effect. This result can be explained by the fact that PCPE-2 binds to BMP-1 through the same domains (CUB1CUB2) as those involved in the interaction with collagen C-propeptides, through partially overlapping binding sites, as demonstrated by competition

and mutagenesis experiments. This results in mutually exclusive interactions of PCPE-2 with the procollagen or the protease and leads to the unusual bell-shaped curve described above, with successive phases of stimulation by PCPE-2 (through procollagen binding), status quo (equilibrium between stimulation and inhibition) and inhibition by PCPE-2 (through protease binding). At the highest concentrations, PCPE-2 works by hijacking BMP-1 to prevent its interaction with procollagen substrates, independently of the presence of PCPE-1 which then becomes totally powerless. It even appears that PCPE-1 potentiates the inhibition of BMP-1 PCP activity by PCPE-2, probably through the saturation of binding sites on procollagens which makes more PCPE-2 molecules available to bind BMP-1. This is reflected by the higher inhibition of procollagen processing observed when both PCPEs are present than when PCPE-2 alone is present.

An important question related to these findings is whether this complex mechanism of regulation of procollagen processing is relevant in vivo. Based on the relatively low expression of *PCOLCE2* in fibroblasts and the absence of major alterations in collagen fibrils, the answer seems to be that PCPE-2 is actually not crucial for proper collagen assembly in homeostatic conditions. However, it cannot be excluded that *PCOLCE2* expression might be induced upon challenge. There is presently no evidence for that, at least in conditions known to trigger high collagen synthesis such as tissue injury, and *PCOLCE2* does not seem to follow the up-regulation trend often displayed by *BMP1* and *PCOLCE* in these conditions. Its protein or RNA level was actually found to be down-regulated in murine models of corneal scarring[14] and lung injury[56,57] and unchanged in a model of cardiac fibrosis[58]. Interestingly also, in skin and kidney, *PCOLCE2* expression seems restricted to specific fibroblast subpopulations with progenitor potential[59,60] rather than with myofibroblast/matrix-producing potential[59,61]. In contrast, *PCOLCE2* is more highly expressed than *PCOLCE* in human macrophages[62,63] and could regulate neutrophil functions[64]. Even though the regulation of *PCOLCE2* expression during injury-driven inflammation remains to be established, these data suggest that PCPE-1 and PCPE-2 might play distinct and non-overlapping roles in tissue repair. In line with this, we have observed here that PCPE-2 can efficiently counteract PCPE-1 activity, potentially leading to conflicting outcomes if both are present simultaneously in the same locations. More intriguingly, our present results do not fit with the findings, described in an earlier report[37], suggesting that *Pcolce2* deficiency protects mice against collagen accumulation and myocardial stiffening in a model of chronic pressure overload. These findings will have to be confirmed and investigated again to determine if they really depend on the direct regulation of BTP-dependent collagen maturation by PCPE-2 or on unrelated mechanisms.

If procollagen processing is not a major target of PCPE-2, other potential functions can nonetheless be proposed for this protein. For example, the proteolytic products resulting from some of the substrates characterized in this study are known to play a role in the control of cell adhesion and migration (TSP-1), angiogenesis (TSP-1, perlecan/endorepellin) and/or BMP and TGF-β activation and bioavailability (chordin, TSP-1, betaglycan). Their regulation by PCPE-2 could thus give mechanistic hints to explain that the latter recently emerged as a signature gene predicting survival in several types of cancers[65–67]. In line with the possible link between PCPE-2 and the regulation of growth factor activity by chordin, TSP-1 and betaglycan, we also made the interesting observation that BMP-2 and TGF-β1 mRNA levels were decreased in the skin of *Pcolce2*-null mice (Supplementary Fig. 1c). Although these results are still preliminary and require further investigation, they could suggest a compensatory down-regulation mechanism following increased BTP-dependent activation of BMP-2 and TGF-β1 in the absence of PCPE-2. Finally, the cleavage of LDLR is strongly inhibited by PCPE-2 suggesting a connection with PCPE-2 relatively well-established roles in the regulation of diet-induced atherosclerosis[39] and adipose tissue expansion[68].

However, these roles were mainly described in mouse models where the LDLR cleavage site that is targeted by BMP-1 in the human protein is not conserved[19]. Other mechanisms are therefore also certainly involved such as a direct interaction of PCPE-2 with other partners (e.g. the scavenger receptor SR-BI, as previously proposed[39,68]) or the cleavage of other BMP-1 substrates linked to lipid metabolism. In this context, a BMP-1 substrate that was unfortunately not available to us but will require further investigation is pro-ApoA1[20]. Its proteolytic maturation by BMP-1 was previously reported to be enhanced in the presence of PCPE-2[69] while our present results would rather predict that it should be inhibited.

In summary, we have shown that with the same combination of two CUB domains sharing more than 50% identity, it is possible to obtain, on one side, a powerful enhancer of one specific activity of a protease family (PCPE-1) and, on the other side, a potent inhibitor of all the other activities of the same protease family (PCPE-2). Future work should now focus on the identification of the molecular determinants which allow this rather amazing specialization. Also, it will be essential to characterize the cell types that are the most relevant in terms of PCPE-2 production and the pathophysiological contexts where PCPE-2 is co-expressed with BTPs in order to better define the scope of the regulatory mechanisms unveiled by the present study.

## Methods

### *Pcolce2*-null mice

*Pcolce2*-null embryos were obtained from the group of Dr. Thomas Boehm at the Max-Planck-Institute of Immunobiology and Epigenetics in Freiburg, Germany. They were generated by replacing a large part of the exon 3 of the *Pcolce2* gene by the IRES-lacZ/neomycin resistance cassette, as described in[38]. Embryos were injected into C57BL/6 females and the offspring were mated with C57BL/6 mice to generate both wild-type (WT) and *Pcolce2*-null (KO) mice. These were housed in the pathogen-free facilities of the University of Freiburg or of the SFR Biosciences (Lyon, France) according to ethical regulations. They were bred under standard conditions (12 h light/dark cycle, 22 ± 2 °C temperature, around 50% humidity, water and food ad libitum). Approximately equal numbers of male and female mice were used in each experiment.

### Transmission electron microscopy

Six week-old KO and WT mice were sacrificed and back skin, heart and tail tendon were carefully dissected and processed for transmission electron microscopy (TEM). Small tissue pieces (around 1 mm³ in size) were fixed in one volume of 4% glutaraldehyde and one volume of 0.2 M sodium cacodylate pH 7.4 at 4 °C. Then, samples were carefully rinsed 3 times at 4 °C and post-fixed with 2% OsO₄ for 1 h at 4 °C before being dehydrated in graded series of ethanol and transferred to propylene oxide. Impregnation was performed with Epon epoxy resin (Electron Microscopy Sciences). Inclusion was obtained by polymerization at 60 °C for 72 h. Ultrathin sections (approximately 70 nm thick) were cut on a UC7 ultramicrotome (Leica), mounted on 200 mesh copper grids coated with poly-L-lysine (Electron Microscopy Sciences), stabilized for 1 day at room temperature and contrasted with uranyl acetate. Sections were examined with a Jeol 1400JEM 120 kV transmission electron microscope (Tokyo, Japan), equipped with a Gatan Orius 600 camera on wide field position and Digital Micrograph software v1.7 (Gatan Inc).

Fibril diameters were analyzed with ImageJ v1.53 on 2-6 TEM images/mouse for skin and tendon and up to 10 images/mouse for heart (4–8 mice/genotype). Diameters were derived from Feret diameters for skin fibrils, cross-sectional areas for heart fibrils and diameters of the smallest inscribed circles for tendons. The scripts of the macros developed to measure fibril diameters in hearts and tendons are available in Supplementary Methods.

## Skin mechanical properties

Skin mechanical properties were measured using uniaxial traction assays as previously described[40]. Briefly, skin from the back of 8-week-old WT and KO mice was depilated, the epidermis was removed using 3.8% ammonium thiocyanate and the remaining dermis was cut into a dog-bone shape along the antero-posterior direction to ensure homogeneous uniaxial tensile load in the central tested portion. The sampled papillary dermis was labelled with a pattern of dots of Indian ink using a soft brush before being attached to the traction device using metallic jaws. The marked surfaces were lit with white light using a LED light (Schott, KL 2500 LED), and images were recorded every 3 s during the test using a digital camera (Allied Vision GX6600) equipped with a telecentric lens (Opto Engineering TC16M036), allowing the observation of the full skin sample. The pattern of Indian ink dots created the macroscopic pattern needed to analyze the experiments with Digital Image Correlation in post treatment using CorrelManuV software (v1.681).

Prior to the experiment, the dimensions of each sample were measured using a caliper. Tensile tests were performed at a strain rate around $0.5\% \, s^{-1}$, without any stops, until rupture of the sample. During each test, the displacement of the grips and the force were recorded every second. The force was divided by the initial section of the sample to obtain the nominal stress. Stretch was determined either through the machine or through the optical measurement, averaged on the whole sample. As reported[70], the optical stretch varies proportionally to the machine stretch (and, in mean, is equal) and we used it as the true one. Finally, four parameters were extracted from the nominal stress vs stretch curve: the tangent modulus of the linear region, the heel region length, the failure stretch and the ultimate tensile stress.

## Protein production

The human PCPE-2 sequence was obtained from the pOTB7-PCPE-2 plasmid purchased from Source BioScience (clone ID 3951739). This sequence contained two mutations (829 G > T and 874 C > A) which were corrected. The PCPE-2 sequence was then cloned into a pJET1.2/blunt vector using the CloneJET PCR Cloning strategy (ThermoFisher). PCPE-2 cDNA was further amplified by PCR with a C-terminal 8His-tag ([Cter-8His]) and inserted into the mammalian expression vector pCEP4, through the Acc65i/BamHI restriction sites. It was also amplified by PCR with an N-terminal 6His-tag ([Nter-6His]) and inserted into the mammalian expression vector pHLsec, in frame with the pHLsec signal sequence through the AgeI/XhoI sites, using the In-Fusion cloning strategy (Ozyme). The PCPE-2 3C construct was designed to insert one HRV 3C-protease cleavage site between the 6His-tag and the beginning of PCPE-2 (giving the following N-terminal sequence after signal peptide removal: ETGHHHHHH**LEVLFQGP**AS) and one HRV 3C-protease cleavage site between amino-acids 270 and 271 of full-length human PCPE-2 (as described in Supplementary Fig. 5c). The corresponding synthetic gene was ordered from Twist Bioscience after codon optimization. The cDNA was digested with AgeI/XhoI and subcloned into linearized pHLsec. The N-terminal fusions of PCPE-2 CUB domains with maltose-binding protein (MBP-CUB1, MBP-CUB1CUB2 and derived mutants) were obtained by subcloning synthetic CUB1 or CUB1CUB2 constructs (obtained from Geneart), with N-terminal HRV 3C-protease cleavage site followed by an NheI restriction site, into the pHLmMBP-1 vector (Addgene plasmid # 72343; deposited by Luca Jovine from the Karolinska Institute, Sweden) through the NotI and XhoI restriction sites (Supplementary Fig. 5e).

To produce the BMP-1 deletion mutants (except BMP-1[cat] described in Supplementary Methods), the sequences corresponding to BMP1[catCUB1] and BMP1[catCUB1CUB2] (including BMP-1 propeptide domain that is cleaved off during protein expression) were amplified by PCR from the previously described pCPE4-BMP-1 plasmid[51] and inserted into the pHLsec plasmid through the AgeI/Acc65i restriction sites. The BMP1[CUB1CUB2] fragment was amplified

(without propeptide) by PCR from the pHLsec-BMP1[catCUB1CUB2] construct, with primers containing NheI and XhoI restriction sites. It was subcloned into the pHLmMBP-1-3C vector generated above and linearized with NheI and XhoI restriction enzymes.

All constructs were checked by Sanger sequencing (Eurofins).

For the production of PCPE-2[Cter-8His], a (HEK) 293-EBNA cell line (Cellulonet, Lyon, France) stably transfected with the pCEP4-PCPE-2 construct was established using Nanofectin (PAA) as transfection agent and 300 µg/ml hygromycin for selection (Merck). Cells were then grown and amplified as previously described for PCPE-1[Cter-8His][51].

Alternatively, transient transfection of 293-T cells (Cellulonet, Lyon, France) was used for the production of PCPE-2[Nter-6His] (the most widely used construct in this study) and PCPE-2 3C. In this case, cells were seeded in HYPERflask or CellSTACK cell culture vessels (Corning) in DMEM AQ medium (Merck) containing 10% of fetal bovine serum (FBS, Gibco or Eurobio) and 1% antibiotic-antimycotic solution (AAS, Merck). At 80–90% confluency, cells were transfected with pHLsec-PCPE-2[Nter-6His] or pHLsec-PCPE-2 3C plasmid, using polyethylenimine (PEI, Merck) as transfection reagent, in DMEM medium with only 2% of FBS and 1% of Non-Essential Amino Acids (Merck). Finally, The MBP-CUB1CUB2 protein and its mutants were produced in (HEK) 293-F cells (ThermoFisher Scientific, ref R79007) grown in suspension in FreeStyle 293 expression medium (Gibco) using sterile flasks (Corning or BD Biosciences) placed on an orbital shaker platform (Eppendorf) rotating at 125 rpm and 37 °C with 8% $CO_2$. On the day of transfection, cells were centrifuged and resuspended at a cell density of $1.5 \, 10^6$ cells/mL in FreeStyle 293 expression medium. For transfection, plasmid DNA and PEI 25 K transfection agent (Polysciences, filtered on 0.2 µm filters) were first diluted separately in 1/20 of the total culture volume of Opti-MEM medium (Gibco) and kept at room temperature for 5 min before mixing. The mixture was further incubated for 15 min before addition to the cells. Finally, 1/5 of the culture volume of fresh medium was added to the cells 24 h after transfection.

Culture media were collected between 3 and 5 days after transfection, centrifuged at 1000 g and mixed with protease inhibitors (0.25 mM Pefabloc (Roth) and 2 mM N-ethylmaleimide (Merck)). The first purification step for PCPE-2 and PCPE-2 3C was on Heparin Sepharose 6 Fast Flow resin (Cytiva) which was prepacked into a column and equilibrated in 20 mM HEPES pH 7.4, 0.15 M NaCl (buffer A). A linear gradient of NaCl was used to elute the protein (from 0.15 to 2 M NaCl) and the PCPE-2 protein was obtained between 0.5 to 0.8 M of NaCl. Imidazole was then added to the pool of PCPE-2-containing fractions to a final concentration of 5 mM before loading the protein solution on Ni-NTA resin (Qiagen). The resin was washed successively with 20 mM HEPES pH 7.4, 0.6 M NaCl containing 20 mM then 50 mM imidazole and the protein was eluted with a gradient of imidazole from 50 to 500 mM. Fractions containing PCPE-2 were pooled, diluted with 20 mM HEPES pH 7.4 to reduce salt concentration down to 0.25 M and loaded on a HiTRAP Heparin-HP column (Cytiva). The protein was eluted with 0.6 M NaCl in 20 mM HEPES pH 7.4, flash-frozen in the same buffer supplemented with 2.5 mM $CaCl_2$ and 0.1% of n-octyl-β-D-glucopyranoside (βOG, Roth) and stored at −80 °C until use. The average yield for PCPE-2 production was 0.5 mg per liter of culture.

MBP-BMP-1[CUB1CUB2], MBP-CUB1CUB2 and its mutants were purified through their His-tag on Ni Sepharose Excel (Cytiva). Cell supernatant was loaded on the resin equilibrated in buffer A. Then, the resin was washed successively with buffer A and buffer A containing 25 mM imidazole and the protein was eluted with a gradient of imidazole from 25 to 500 mM. Fractions containing the protein were pooled and submitted to size exclusion chromatography on a Superdex 200 increase (Cytiva) with 20 mM HEPES pH 7.4, 0.5 M NaCl, 2.5 mM $CaCl_2$ as equilibration buffer.

BMP-1[CUB1CUB2], CUB1CUB2 (from PCPE-2), NTR, CUB1CUB2* and its mutants were generated by cleavage of the corresponding

MBP-BMP-1[CUB1CUB2], PCPE-2 3C or MBP-CUB1CUB2 proteins purified as above until elution from the Ni-NTA column. HRV 3C-protease (with a GST- or His-tag; kind gifts of the Protein Science Facility of the SFR Biosciences, Lyon, France) was added to semi-purified proteins in a 1/30 w/w ratio and the mixture was incubated for 1 h 30 at room temperature, followed by dialysis overnight at 4 °C in a 7 kDa Slide-a-Lyzer (ThermoFisher) or by desalting on ZebaSpin columns (ThermoFisher) to eliminate imidazole. Purification of cleavage mixtures was achieved by purification on Ni-NTA agarose equilibrated in 20 mM HEPES pH 7.4, 0.3 M NaCl, 0.1% βOG followed by desalting for BMP-1[CUB1CUB2] and CUB1CUB2*. To separate CUB1CUB2 and NTR generated from PCPE-2 3C cleavage, an additional step on HiTRAP Heparin-HP column equilibrated in the same buffer was required. Finally, MBP-CUB1 was first purified on a Dextrin Sepharose High Performance resin (Cytiva) equilibrated in 20 mM HEPES pH 7.4, 0.5 M NaCl and eluted with the same buffer containing 10 mM maltose. After HRV 3C cleavage, the cleaved CUB1 protein was retrieved in the flow-through of a second step of Dextrin Sepharose High Performance chromatography. Proteins were stored in 20 mM HEPES pH 7.4, 0.5 M NaCl, 2.5 mM CaCl$_2$ and 0.1% βOG at −80 °C (after flash-freezing) until use.

Protein concentrations were determined from the optical density at 280 nm with a Nanodrop 2000 (Thermo Scientific), using absorption coefficients computed with the Expasy ProtParam tool, or from the Bradford assay (Coomassie Plus Assay reagent, Pierce).

Further information about recombinant proteins is provided in Supplementary Methods.

### Cleavage assays

All cleavage assays with protein substrates were performed at 37 °C in 50 mM HEPES pH 7.4, 5 mM CaCl$_2$ and 0.02% βOG. The NaCl concentration could vary between assays in the range 0.15 M−0.25 M but was kept consistent between comparable conditions. All other concentrations and incubation times were as indicated in figure legends. Analysis was by SDS-PAGE with 4–20% polyacrylamide Criterion gels (BioRad), unless otherwise stated, and Coomassie Blue, Instant Blue (Euromedex) or Sypro Ruby (Merck) staining. For Sypro Ruby staining, a maximum amount of 250 ng of substrate/lane was used to stay in the linear range and gels were analyzed with a Typhoon FLA9500 scanner (blue laser, LPG emission filter; FLA9500 software, Cytiva). Uncropped gels are available in Supplementary Fig. 16 or as a Source Data file.

When applicable, quantification of protein band intensities was made with the ImageQuant TL software v8.2 (Cytiva). Enhancement factors for mini-procollagens and CPIII-Long were calculated as the following ratio: [intensity of Intermediate 1 (1 chain cleaved) + 2 x intensity of Intermediate 2 (2 chains cleaved) + 3 x intensity of C-propeptide] / [3 x (sum of intensities of all cleavage products + procollagen)] and normalized to the BMP-1 alone condition.

Cleavage of the fluorogenic peptide Mca-YVADAPK(Dnp)-OH (20 μM, Covalab) was monitored in the same buffer conditions[26]. Fluorescence was recorded for 20 min (excitation: 320 nm; emission: 405 nm) in a Tecan Infinite M1000 fluorimeter equipped with the iControl software v1.10 and the slope of the curve was used to derive BMP-1 activity. The apparent inhibition constant ($K_i^{app}$) value of PCPE-2 was derived from the data of Fig. 4b using the Morrison equation for tight-binding inhibitors[49] in GraphPad Prism (Eq. 1):

$$\frac{v_i}{v_o} = 1 - k * \frac{([E]+[I]+K_{iapp}) - \sqrt{([E]+[I]+K_{iapp})^2 - 4[E][I]}}{2[E]} \quad (1)$$

where $v_i$ is the measured velocity in the presence of inhibitor, $v_0$ the velocity in the absence of inhibitor, [E] the total enzyme concentration, [I] the added inhibitor concentration, $K_i^{app}$ the apparent inhibition constant, and k a constant reflecting partial inhibition.

### SPR experiments

Surface plasmon resonance experiments were run with a Biacore T200 apparatus (Cytiva) equipped with the Biacore T200 control software (v3.2.1). Protein ligands were covalently immobilized on Series S sensor chips CM5 (Cytiva) by amine coupling chemistry using the reagents included in the Amine coupling kit (Cytiva). Ligands were diluted in 10 mM sodium acetate pH 5 for mini-procollagen III, C-propeptide III and BMP-1 or in 10 mM HEPES pH 7.4 or 8.0 for PCPE-2. SPR signals were recorded simultaneously on a control channel where the same immobilization procedure was applied, except for the presence of protein ligands. Soluble analytes were injected at 30 or 50 μl/min at 25 °C after dilution in running buffer (10 mM HEPES pH 7.4, 0.15 M NaCl, 5 mM CaCl$_2$ and 0.05% P20) for 90 or 120 s, either in high performance or dual inject modes. In co-injection experiments, the first binding partner was injected for 60 s followed by the injection of the two co-injected partners, as indicated, for another 60 s. In all cases, regeneration was achieved with 2 M guanidinium chloride. The proteins used for these experiments were the same as for activity assays except for PCPE-1 which was used as the native form with no His tag in SPR analyses, as described[44]. BMP-1 hydroxamate inhibitor (UK 383,367) was from Merck. Finally, best fits of the data in kinetic and steady-state modes were obtained with the Biacore T200 evaluation software (v3.2.1). Both the 1:1 binding (A + B ↔ AB) and heterogenous ligand (A + B ↔ AB and A + B′ ↔ AB′ with B and B′ representing two different presentations/conformations of the immobilized ligand) models were tested and when a significantly better fit of the data was obtained with the heterogenous complex model, the latter was selected to compute kinetic and steady-state constants.

### Circular dichroism

CD measurements were carried out using 1-mm path length quartz cells in an Applied Photophysics Chirascan instrument, calibrated with aqueous d-10-camphorsulfonic acid. Proteins (0.2 mg/ml) were analyzed at 25 °C in 10 mM Tris·HCl pH 7.4, 0.15 M NaCl, 5 mM CaCl$_2$. The spectra were measured with a wavelength increment of 0.2 nm, an integration time of 1 s and a bandpass of 1 nm. Spectra were processed, baseline-corrected, smoothed and converted with the Chirascan software. Spectral units were expressed as the mean molar ellipticity per residue.

### Cell culture

293-EBNA cells stably transfected with the empty pCEP4 vector, the pCEP4-sizzled vector or the pCEP4-PCPE-2[Cter-8His] vector were available from previous studies[5,26] or obtained as described above. All 293-EBNA cell lines were grown in DMEM (Merck) with 10% FBS and 300 μg/ml hygromycin. At confluency, cells were kept for 20 h in serum-free medium with no antibiotic and the supernatant was collected, supplemented with protease inhibitors (0.25 mM Pefabloc, 2 mM NEM, 2 mM EDTA), centrifuged and stored at −80 °C. The cell lysate was prepared in RIPA buffer (50 mM Tris pH 8, 0.15 M NaCl, 1% NP-40, 1% SDS, 0.5% sodium deoxycholate) containing protease inhibitors (Complete, Roche) and then centrifuged and stored at −80 °C until use. Protein concentrations in cell extracts were measured with the Bradford assay using the Coomassie Plus Assay reagent (Pierce) and BSA as a standard.

Murine fibroblasts (NMF) were isolated from the back skin of C57BL/6 mouse pups by first digesting skin with 1 mg/ml dispase at 4 °C overnight and then mechanically stripping the epidermis from the dermis using forceps. The dermis was collected and processed for fibroblast isolation by mincing it with scissors and incubating it with 500 U/ml collagenase I at 37 °C for 1 h. The digested slurry was passed through a 70 μm cell strainer, cell pelleted with centrifugation, dissolved in DMEM/F12 (ThermoFisher Scientific), 10% FBS (Merck), 2 mM L-glutamine (Merck) and 1% AAS and seeded in tissue culture flasks. The cells were maintained in complete DMEM/F12 and cells in passage

1 or 2 used for the analyses. Primary human fibroblasts were isolated by outgrowths of explant cultures from breast and abdominal skin of healthy female donors undergoing plastic surgery and aged 25–35 years by the Cell and Tissue Bank of Edouard Herriot Hospital (Lyon, France) or the Department of Dermatology, Medical Center of the University of Freiburg (Freiburg, Germany). Cells were harvested in agreement with the French and German ethical regulations (permanent authorization of the French Ministry of Higher Education, Research and Innovation AC-2019-3476 and ethics committee of the University of Freiburg approval no. 318/18). The study was performed in agreement with the principles of the Declaration of Helsinki and donors gave written informed consent for use of the materials for research. Human fibroblasts were cultured in complete DMEM/F12 or DMEM AQ™ medium with 10% FBS and 1% AAS. Mouse and human fibroblasts were passaged and detached using trypsin-EDTA (PAN Biotech or Merck).

## Immunoblotting

Proteins were separated by SDS-PAGE in 25 mM Tris pH 8.5, 0.2 M Glycine, 0.1% SDS buffer (Euromedex). Proteins were then electroblotted for 2 h onto polyvinylidene difluoride (PVDF; 0.45 μm) membranes (Millipore) in 10 mM CAPS (N-cyclohexyl-3-aminopropanesulfonic acid) pH 11 with 10% ethanol. If quantification of total protein amount was required, Stain-Free activation was performed on the acrylamide gel for 30 s using a Fusion FX system (Vilber Lourmat) prior to electroblotting and transferred proteins were visualized using Stain-Free detection on the PVDF membrane after electroblotting. Membranes were then blocked with 10% skim milk in PBS for 1 h and incubated overnight with primary antibodies diluted into blocking buffer solution (5% skim milk in PBS with 0.05% Tween 20) at 4 °C (see details of antibodies below). After three washes in PBS containing 0.05% Tween 20, horseradish peroxidase (HRP)-coupled secondary antibodies were added for 1 h at room temperature. Proteins of interest were detected by chemiluminescence with Amersham ECL Select Western blotting detection reagent (Cytiva) using the Fusion FX camera system equipped with the FusionCapt Advance FX7 16.16b software (Vilber Lourmat). When applicable, quantification of band intensities was performed with the FusionCapt Advance FX7 16.16b software or the Image-Quant TL software v8.2 (Cytiva). Uncropped immunoblots are available in Supplementary Fig. 16 or as a Source Data file.

Human and mouse C-propeptides of procollagen I were detected with the LF41 antibody[71] which was a kind gift of Dr. Larry W. Fisher (NIH, Bethesda, USA; dilution 1/1000). LDLR was detected with AF2148 (dilution 1/2000), human PCPE-1 with AF2627 (dilution 1/1000), mouse PCPE-1 with AF2239 (dilution 1/1000) and His-tagged proteins with MAB050 (dilution 1/5000), all from BioTechne. An anti-PCPE-2 antibody was generated in rabbits against a synthetic peptide corresponding to amino-acids 201-230 of human PCPE-2 (DVERDNYCRY-DYVAVFNGGEVNDARRIGKY) by the Covalab company (France). Rabbit serum was first purified against the immunogen and then against human PCPE-1 to remove all potential antibodies which could cross-react with PCPE-1. The purified antibody was used at a dilution of 1/1500. Horse anti-mouse secondary antibody (# 7076 S) and goat anti-rabbit secondary antibody (# 7074 S) were from Cell Signaling. Donkey anti-goat secondary antibody (DkxGt-003-DHRPX) was from ImmunoReagents. All secondary antibodies were used at a dilution of 1/10,000.

## RNA isolation and quantitative real-time PCR

Total RNA from primary cells was isolated using the RNeasy Plus Mini kit (Qiagen) or the Nucleospin RNA kit (Macherey-Nagel) according to the manufacturer's instructions. RNA from mouse skin was extracted using the RNeasy Fibrous Tissue Mini kit (Qiagen). Quality and quantity of the isolated RNA were determined with a Nanodrop 2000. RNA

(500 ng) was then reverse-transcribed to cDNA using the First Strand cDNA Synthesis Kit (ThermoFisher Scientific) or the PrimeScript RT-PCR kit (Takara). Quantitative real-time PCR (qRT-PCR) analyses were performed with a CFX96 Real-Time system equipped with the CFX Manager Software software (BioRad) or a Rotor-Gene Q system equipped with the Rotor-Gene Q software v2.3.5 (Qiagen). qRT-PCR reactions were made using the iQ SYBR Green Supermix (Bio-Rad) or the FastStart Universal SYBR Master mix (Roche Applied Science) according to the manufacturers' recommendations with the primers described in Supplementary Table 1 (amplification efficiency > 90%). PCR conditions were as follows: initial denaturation at 95 °C for 10 min then 50 amplification cycles with denaturation at 95 °C for 10 s and annealing at 60 °C for 20 s. Fluorescence was measured at the end of the annealing step of each cycle and quantification cycles (Cq) were determined for each gene and condition. At the end of the amplification, a melting curve was recorded by heating at 0.5 °C/s from 70 °C to 94 °C. Relative quantification was performed from three technical replicates using the $2^{-\Delta\Delta Cq}$ method, which allows the comparison of two conditions after normalization with the reference genes (*GAPDH* for human cells and *Gapdh* or *Rpl13a* for mouse cells).

## PCPE-2 structure modeling and sequence alignments

Sequence alignments were performed using Clustal Omega (via the Uniprot web server) and rendering using Clustal Omega or ESPript 3.0 (https://espript.ibcp.fr), based on Uniprot sequences Q15113 for PCPE-1 and Q9UKZ9 for PCPE-2 and on structural information extracted from PDB entry 6FZV[34]. Protein structures were drawn using UCSF Chimera (https://www.cgl.ucsf.edu/chimera) or ChimeraX (http://www.rbvi.ucsf.edu/chimerax). YASARA (http://www.yasara.org) was used for homology modeling of PCPE-2 and computing of rmsd.

## Statistical analysis

Means, medians, ranges and standard deviations were calculated with Excel 2016 or 2019. Other statistical analyses were performed with GraphPad Prism 8 or 9. Using a 5% threshold, we first verified the normality of the distribution using the Shapiro-Wilk test as well as the equality of variance with the Fisher test. If these tests were passed successfully, the significance of the mean difference was assessed using an unpaired two-sided t-test. In other situations, the test used was as indicated in figure legends. For multiple comparisons, one-way ANOVA for paired data (or mixed-effects analysis in case of missing values) was applied. Corrections, post-tests and p-values were as indicated in figure legends or above graphs.

## Reporting summary

Further information on research design is available in the Nature Portfolio Reporting Summary linked to this article.

## Data availability

The data generated in this study are provided in the Supplementary Information and as a Source Data file. The structural data used in this study are available in the Protein Data Bank under accession code 6FZV. Plasmids for recombinant proteins are available upon request. Source data are provided with this paper.

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

## Acknowledgements

We thank the staff of the PBES animal facility (SFR Biosciences, Lyon, France, UAR3444/CNRS, US8/INSERM, ENS de Lyon, UCBL) for breeding and housing the mice. We are grateful to Jacques Brocard and Christophe Chamot (PLATIM facility, Lyon Multiscale Imaging Center and SFR Biosciences) and to Denis Ressnikoff and Bruno Chapuis (CIQLE facility, Lyon Multiscale Imaging Center and SFR Santé-Lyon Est) for their help with the quantitative analysis of collagen fibrils. We acknowledge the contribution of Virginie Gueguen-Chaignon from the Protein Science Facility of the SFR Biosciences for circular dichroism experiments and for the kind gift of the HRV 3C proteases. We also thank Fernando Lopez-Casillas (Universidad Nacional Autonoma de Mexico, Instituto de Fisiologia Celular, Mexico) for providing recombinant betaglycan and David Hulmes (LBTI, CNRS-University of Lyon) for critical reading of the manuscript. Finally, we acknowledge the contribution of Alexis Cellier, Paulin Arès, Julie Devcic and Dana Tatah (LBTI, CNRS-University of Lyon) to the optimization of the production of PCPE-2 and BMP-1-derived proteins. This work was supported by the CNRS, the University of Lyon, ANR grants 18-CE92-0035-01 and 21-CE11-0020-01 to CM, ANR grant 17-CE14-0033 to SVLG, DFG grants SFB850, project B11 and NY90/5–1 to AN and DFG grant SFB877, project A9 to CBP.

## Author contributions

S.V.L.G., A.T., L.B.T., A.N. and C.M. designed research. S.V.L.G., A.T., M.N., C.D., J.B., M.D., P.L., C.P., L.E., S.K., N.T., E.B., N.M., E.E.C., J.M.A., A.N. and C.M. performed research. C.F.G., F.R. and C.B.P. provided reagents. S.V.L.G., A.T., M.N., C.D., J.B., M.D., P.L., C.P., L.E., N.T., E.B., C.B.P., J.M.A., L.B.T., A.N. and C.M. analyzed the data. S.V.L.G., A.N. and C.M. wrote the paper.

## Competing interests

The authors declare the following competing interests: S.V.L.G. and C.M. have filed a patent including PCPE-2 mutants described in Fig. 5

(Applicant: CNRS, UCBL; Inventors: Sandrine Vadon-Le Goff and Catherine Moali; Application number: EP23306041; Status of application: pending). Other authors declare no competing interests.
