## [Peer Review File · Nature Communications]

Identification of PCPE-2 as the endogenous specific inhibitor of human BMP-1/tolloid-like proteinases

Editorial Note: Parts of this Peer Review File have been redacted as indicated to maintain the confidentiality of unpublished data.REVIEWER COMMENTS

Reviewer #1 (Remarks to the Author):

This paper describes the unusual ability of C-proteinase enhancer-2, PCPE-2 to modify the activity of the mammalian tolloid family members, BTPs, notably BMP-1, by binding outside the catalytic domain of these enzymes, probably by interaction with their CUB domains. The authors compare PCPE-2 to the well described PCPE-1 form, of similar domain structure but significantly different underlying primary sequence. The data indicate that PCPE-2 could play an important role in regulating the non-collagenolytic functions of BTPs, contrary to the (sparse) published literature on this topic. As current observations from several groups have found very few phenotypic changes in PCPE-2 null mice and very low levels of tissue expression outside the heart, its importance remains unclear. Overall this protein remains something of an enigma and will require further analysis of its potential roles in mammalian biology. The target proteases that may be regulated by PCPE-2 are themselves in need of further investigation in terms of their cell biology, notably in relation to lipid metabolism.

The manuscript is well written and the experiments carefully conducted with substantial relevant supplementary information. It goes some way towards clarifying the existing conflicting literature on PCPE-2, particularly in relation to collagen maturation. Potential new avenues of investigation have been opened although they are not addressed in detail in this submission.

More specific comments:

1. Abstract: The description of PCPE-2 as a 'potent' inhibitor is patently incorrect, based on the data presented. It is intriguing in that it is unlike most mammalian protease inhibitors, with no active site interactions and the requirement of excessive molar ratios to achieve substantial inhibition (see below). Its capacity to fine tune BTP activities needs to be demonstrated in cell based systems to support this thesis.
2. Results p.5: Was PCPE-2 expression in other isolated cells of potential relevance, e.g. macrophages, c.f. refs 60,61, analysed? Given that this study essentially rules out PCPE-2 as a regulator of fibrillar collagen processing, the role in growth factor etc. processing becomes rather important and the study of relevant cell types might be informative?
3. Results p.9: The kinetic studies of PCPE-2 inhibition of BMP-1 using a short peptide substrate are confusing in terms of explaining the possible biological relevance of PCPE-2 regulation of BTP's in biological systems. It is clear in Fig. 4a that large molar ratios of PCPE-2 to enzyme are required to see significant inhibition and that close to 30% enzyme activity remains even at 30x molar excess. This does not really reflect a 'potent' inhibitor (Abstract and Discussion). The calculation of K_i is misleading as this will depend on substrate concentration and is something of a biochemical artefact. Comparing Figs

4a and b the data do not agree if 7nM BMP-1 was used in both studies. In 4a 7nM PCPE-2 (equimolar) gives ~25% inhibition and in 4b ~40% inhibition is recorded.

4. Fig.7, summarising the observations is not especially informative. More molecular understanding of the PCPEs is necessary and, above all, more biological evidence of the role of PCPE-2.

Reviewer #2 (Remarks to the Author):

Dear Editor,

Please, find my comments regarding the reviewing of the manuscript from Vadon-Le Goff et al. (ref NCOMMS-22-34298), entitled "Identification of PCPE-2 as the first endogenous specific inhibitor of human BMP-1/tolloid-like proteinases".

In this study, the authors investigated the regulatory activity of procollagen C-proteinase enhancer-2 (PCPE-2) towards BMP-1/tolloid-like proteinases (BTPs), and found that PCPE-2, contrary to PCPE-1 that is a well described enhancer of BTPs, can inhibit BTP protease activity. Using BMP1 as main model, the authors, showed that depending on its concentration, PCPE-2 could exhibit PCPE-1 like enhancing activity, have no effect, or inhibit BMP1 through a direct interaction involving the non-catalytic domain of the protease and the CUB domains of regulatory protein. The authors also reported a similar stimulatory/inhibitory activity of PCPE-2 towards mTLL-1, another member of the BTP family, and that PCPE-2 could also inhibit the PCP activity of Meprin α / β metalloproteinases.

Overall, data are sound and the presented work is of quality. Findings are of particular interest as no naturally occurring inhibitor of BTPs had been identified so far, and they encourage a change of paradigm by challenging the historically alleged PTB enhancing activity of PCPE-2.

Although some major issues remain unaddressed (underlying mechanisms of interaction/inhibition, link of BTP inhibitory properties of PCP-2 with a precise physiological function or pathological condition...), I believe these new findings of highly significant interest to justify publication in Nature communications. I have however a number of concerns that I would wish the authors to address:

Main scientific comments:

1- The authors claim that PCPE-2 is a specific inhibitor of BMP-1/tolloid-like proteinases, based on their analysis of BMP1 and mTLL-1. This of course raises the question of the other members of the family, i.e. mTLD and TLL-2. Demonstrating restricted or pan-inhibitory properties of PCPE-2 for the whole metalloproteinase tolloid family would be a significant plus to the study.

2. One of the striking findings of the present study is the potential antagonist activity of highly homologous PCPE-1 and PCPE-2, the latter being able to bind and block BMP1. I concede to the authors that clarifying the precise structural basis of PCPE-2/BMP1 interaction may fall outside the scope of the present study. However, the manuscript would gain in impact if the authors could provide further insights into the underlying mechanism. The authors show that full length PCPE-2 and PCPE-2 isolated CUB1CUB2 domains interact with BMP-1, but not full length PCPE-1. As AlphaFold models for PCPE-1 and PCPE-2 suggest distinct domain spatial distribution, I believe it would be interesting to analyze the potential interaction of BMP-1 with the isolated CUB1CUB2 domains of PCPE-1. This would provide a relevant information about a potential indirect role of the NRT domain, which may not be directly involved in the interaction, but could prevent access of BMP1 to CUB1CUB2 domains in PCPE-1.

3- In this manuscript, most of the binding analyses have been achieved using SPR. Although data are broadly convincing, it should be noted that in some instances, authors chose to fit their binding data using the "Heterogeneous ligand model", (eg in figure 4f and 4g, also in figure supp7b). I am well aware of the difficulties of fitting interaction data when using complex biomolecules in SPR, but this is less expected from a ligand such as BMP-1. I believe that this decision should be justified (in mat&meth). Was the 1:1 model completely inappropriate, or discarded because of poor χ^2 (but coherent kinetic data and K_d)?

In addition, in Supp figure 7B, the range of concentrations injected (0.78-50 nM) are below the reported K_d (80 nM using the steady state fit), which is not suitable for precise affinity determination. The risk of inaccuracy should be brought to the attention of the reader.

Other minor issues

1. The study shows many SPR studies using different ligand/analyte pairs, which makes it sometimes difficult to follow. The manuscript would gain in clarity if the authors could state the ligand used on the figure, for each SPR analysis.

2. In Supp. Fig 5, the authors observe 2 bands for the CUB1/CUB2 domain that they claim to be 2 distinct /cub2 O-glycosylated forms. Have these O-glycosylations been described before? Did the authors confirmed this by O-Glycosidase digestion?

Reviewer #3 (Remarks to the Author):

The authors present a very interesting study with potentially strong clinical significance for the field. However, in my opinion it needs to be better clarified how the difference between PCPE1 and PCPE2 in their CUB domains directs substrate specificity.

Major points:

1. Quite often, protease substrates, inhibitors and enhancers of protease activity create feedback loops that regulate gene expression. To that end, it would be important and interesting if the authors could investigate the mRNA expression profile of Bmp1, tll-1/2, PCPE1, betaglycan, chordin, Bmp2, Bmp4, LDLR receptor in WT and Pcolce2-null mice.

2. Ref. 36 showed that PCPE2 is a procollagen C-proteinase (BMP-1 and mTLL-1) enhancer and that PCPE1 and -2 compete with procollagen C-proteinases for collagen I and II binding.

>In ref. 36 recombinant protein constructs with similar designs were compared to each other. In the present study sensitive activity tests with N- and C-terminally tagged protein constructs of PCPE1 and PCPE2 were compared to each other. Can the authors demonstrate that placement of the tags at different positions does not influence activity?

>also often times not the appropriate controls were used: for instance in several Figs. CUB or NTR domains of PCPE2 were compared to PCPE1 full length and not to CUB and NTR domains of PCPE1.

Can the authors reproduce the same results with the appropriate controls next to each other?

It is possible that expression of separate domains lead to conformational changes which lead to different functional properties.

Can the authors provide CD measurements to assure that the conformation of CUB and NTR domains are preserved?

3. The most interesting aspect of the paper is that apparently CUB domains of PCPE1 and PCPE2 are responsible for their different behavior, although their conformation seems similar (Fig2D). Can the

authors pinpoint the exact structural features present in the CUB domains of PCPE2 that lead to inhibition instead of enhancement of processing as already shown for the CUB domains of PCPE1? In Fig. 2D a superposition is shown. However, an alphafold model or CD measurements of CUB domains from both PCPE1 and PCPE2 together with the sequence alignments shown in suppl. Fig6 would allow the authors to identify the crucial residues responsible for the observed differences in functional behavior. If indeed a stronger affinity of the PCPE-2 CUB domains to BMP-1 is the reason for the BMP-1 inhibition of processing, then site-directed mutagenesis should be employed to neutralize the interaction of PCPE-2 to BMP-1. The generated PCPE-2 mutant should be assessed for inhibition or enhancement of BMP-1 proteolytic processing. CD spectra measurements should be carried out to ensure that the introduced mutations do not affect PCPE2 folding but specifically loosen its interaction with BMP-1. The authors have gathered extensive experience about the structure-function relationship of CUB domains of PCPE1 as evidenced by ref.2 of the supplementary references and have the knowledge and expertise to carry out these analyses.

4. Conceptionally, it should be made better clear to the audience why PCPE2 acts as an inhibitor. The only logical explanation is that it interacts either with different affinities to different substrates or as the authors pointed out it is that the kinetics of the tripartite complex formation of the substrates and BTPs are significantly altered. In my opinion more work needs to be invested to better answer this question. Affinities of CUB domains of PCPE1 and PCPE2 with different substrates should be determined. Also tripartite complex formation should be exemplarily shown for some critical substrates, such as procollagen II versus procollagen III.

Minor: In the working model also the specific functions of the different functional domains should be illustrated

Responses to reviewers' comments in blue

Reviewer #1 (Remarks to the Author):

This paper describes the unusual ability of C-proteinase enhancer-2, PCPE-2 to modify the activity of the mammalian tolloid family members, BTPs, notably BMP-1, by binding outside the catalytic domain of these enzymes, probably by interaction with their CUB domains. The authors compare PCPE-2 to the well described PCPE-1 form, of similar domain structure but significantly different underlying primary sequence. The data indicate that PCPE-2 could play an important role in regulating the non-collagenolytic functions of BTPs, contrary to the (sparse) published literature on this topic. As current observations from several groups have found very few phenotypic changes in PCPE-2 null mice and very low levels of tissue expression outside the heart, its importance remains unclear. Overall this protein remains something of an enigma and will require further analysis of its potential roles in mammalian biology. The target proteases that may be regulated by PCPE-2 are themselves in need of further investigation in terms of their cell biology, notably in relation to lipid metabolism.

The manuscript is well written and the experiments carefully conducted with substantial relevant supplementary information. It goes some way towards clarifying the existing conflicting literature on PCPE-2, particularly in relation to collagen maturation. Potential new avenues of investigation have been opened although they are not addressed in detail in this submission.

We thank the reviewer for this positive overall evaluation of the manuscript. We hope that our study will indeed boost the investigation of novel functions for PCPE-2 and BTPs which remain largely under-explored.

More specific comments:

1. Abstract: The description of PCPE-2 as a 'potent' inhibitor is patently incorrect, based on the data presented. It is intriguing in that it is unlike most mammalian protease inhibitors, with no active site interactions and the requirement of excessive molar ratios to achieve substantial inhibition (see below). Its capacity to fine tune BTP activities needs to be demonstrated in cell based systems to support this thesis.

A detailed response in support of the use of "potent" is given below.

We would also like to make the point that "no active site interaction" does not mean no interaction between PCPE-2 and the catalytic domain of BMP-1. What is clear from the data presented in new Fig. 4 is that PCPE-2 requires BMP-1 auxiliary domains (especially CUB1 and CUB2) to bind and inhibit the protease. However, this does not exclude cooperative interactions where prior binding of PCPE-2 to the CUB domains of BMP-1 would help to recruit the catalytic domain in order to promote the allosteric control of protease activity. Our previous experience with CUB domains has shown that they are very prone to this type of cooperative interactions (Kronenberg et al. 2009; DOI : 10.1074/jbc.M109.046128) and the fact that the cleavage of a short peptide, probably having no interaction with non-catalytic domains, is affected by PCPE-2 is in favor of a direct effect on the catalytic domain.

Regarding the ability of PCPE-2 to fine tune BTP activities in cell-based assays, the data shown in Fig. 3g clearly indicate that transfected PCPE-2 can also inhibit the processing of endogenous LDLR by endogenous BMP-1 in a cellular system.

2. Results p.5: Was PCPE-2 expression in other isolated cells of potential relevance, e.g. macrophages, c.f. refs 60,61, analysed? Given that this study essentially rules out PCPE-2 as a regulator of fibrillar collagen processing, the role in growth factor etc. processing becomes rather important and the study of relevant cell types might be informative?

We have started to analyze PCPE-2 pattern of expression in more detail, both in terms of cell types and of subcellular localization. PCPE-2 is expressed at the RNA level in macrophages. [redacted] However, the role of BMP-1 in these cells being completely unknown, it is presently difficult to select a relevant pathway/physiological setting to look at and we think that it falls beyond the scope of the present study. However, we already obtained the interesting result that the expression of at least two of the growth factors known to be regulated by BTPs (BMP-2 and TGF- β) is down-regulated in the skin of Pcolce2-null mice (Supplementary Fig. 1c).

3. Results p.9: The kinetic studies of PCPE-2 inhibition of BMP-1 using a short peptide substrate are confusing in terms of explaining the possible biological relevance of PCPE-2 regulation of BTP's in biological systems. It is clear in Fig. 4a that large molar ratios of PCPE-2 to enzyme are required to see significant inhibition and that close to 30% enzyme activity remains even at 30x molar excess. This does not really reflect a 'potent' inhibitor (Abstract and Discussion). The calculation of K_{iapp} is misleading as this will depend on substrate concentration and is something of a biochemical artefact.

As indicated by the reviewer, complete inhibition of BMP-1 by PCPE-2 cannot be achieved for short peptide substrates while it is observed for physiological substrates both *in vitro* and *in cellulo* (e.g. Fig. 3a and f, g). The difference between peptide and protein substrates probably comes from the contribution of exosites (in non-catalytic domains) which also plays a role in binding physiological substrates and inhibitors. However, even if not complete, maximum inhibition is reached around 30 nM PCPE-2 (4-molar excess to BMP-1) for the fluorogenic peptide. Also, the use of such peptides remains by far the easiest way to evaluate IC_{50} and K_i as it is currently the only way to measure BMP-1 activity in real-time to ascertain linearity. Both IC_{50} and K_{iapp} are now indicated in Fig. 4b.

About the use of "potent" to describe inhibition of BMP-1 by PCPE-2, it refers to the fact that PCPE-2 can be defined as a tight-binding inhibitor acting in nearly stoichiometric fashion (see ref 49 : "The simplest determination that tight binding inhibition is occurring comes from measurement of the dose-response curve for inhibition. An IC_{50} value obtained from this treatment of the data that is similar to the concentration of total enzyme in the sample (i.e., within a factor of 10) is a good indication that the inhibitor is of the tight binding type"). Here the calculated IC_{50} is 6.9 ± 0.9 nM and is almost identical to enzyme concentration (7 nM) showing that PCPE-2 is a tight-binding inhibitor. Finally, complete inhibition with physiological substrates can be achieved with PCPE-2:BMP-1 molar ratios below 4 (see the example of pN α 1(V) in Fig. 3a), also indicating that PCPE-2 is a potent inhibitor of BMP-1 that compares well with other endogenous inhibitors of metalloproteinases (e.g. Fetuin B/Merin beta : $K_i = 7$ nM (Karmilin et al. 2019; DOI : 10.1038/s41598-018-37024-5); TIMP-3/ADAMTS4 : $K_{iapp} = 3.3$ nM (Kashiwagi et al. 2001 ; DOI : 10.1074/jbc.C000848200); TIMP-1/MMP-9 : $K_i = 8.5$ nM (Olson et al. 1997; DOI : 10.1074/jbc.272.47.29975).

Comparing Figs 4a and b the data do not agree if 7nM BMP-1 was used in both studies. In 4a 7nM PCPE-2 (equimolar) gives ~25% inhibition and in 4b ~40% inhibition is recorded.

We thank the reviewer for pointing this discrepancy. It made us realize that the molar ratio of PCPE-2 to BMP-1 was slightly wrong on the previous figure and appropriate concentrations are now given in the legend of Fig. 4a (PCPE-2 is 5 nM instead of 7 nM as previously indicated). With 5 nM of PCPE-2, we observe 22 ± 7 % inhibition in Fig. 4a to be compared to 20 ± 6 % inhibition in Fig. 4b.

4. Fig.7, summarising the observations is not especially informative. More molecular understanding of the PCPEs is necessary and, above all, more biological evidence of the role of PCPE-2.

We think that it is important to keep a summary figure due to the complexity of the mechanisms involved and to illustrate the fine-tuning of BTP activities by PCPEs. The model has been updated in response to the comments of reviewer 3 and in light of the new molecular information provided in the revised manuscript regarding the implication of the various domains in PCPE-2 and BMP-1. We hope that it will thus appear more informative to all reviewers.

Reviewer #2 (Remarks to the Author):

Dear Editor,

Please, find my comments regarding the reviewing of the manuscript from Vadon-Le Goff et al. (ref NCOMMS-22-34298), entitled "Identification of PCPE-2 as the first endogenous specific inhibitor of human BMP-1/tolloid-like proteinases".

In this study, the authors investigated the regulatory activity of procollagen C-proteinase enhancer-2 (PCPE-2) towards BMP-1/tolloid-like proteinases (BTPs), and found that PCPE-2, contrary to PCPE-1 that is a well described enhancer of BTPs, can inhibit BTP protease activity. Using BMP1 as main model, the authors, showed that depending on its concentration, PCPE-2 could exhibit PCPE-1 like enhancing activity, have no effect, or inhibit BMP1 through a direct interaction involving the non-catalytic domain of the protease and the CUB domains of regulatory protein. The authors also reported a similar stimulatory/inhibitory activity of PCPE-2 towards mTLL-1, another member of the BTP family, and that PCPE-2 could also inhibit the PCP activity of Meprin α / β metalloproteinases.

Overall, data are sound and the presented work is of quality. Findings are of particular interest as no naturally occurring inhibitor of BTPs had been identified so far, and they encourage a change of paradigm by challenging the historically alleged PTB enhancing activity of PCPE-2.

Although some major issues remain unaddressed (underlying mechanisms of interaction/inhibition, link of BTP inhibitory properties of PCP-2 with a precise physiological function or pathological condition...), I believe these new findings of highly significant interest to justify publication in Nature communications.

We thank the reviewer for the very positive evaluation of our manuscript and hope that the new data described in the revised version will address some of the points which were previously unclear.

I have however a number of concerns that I would wish the authors to address:

Main scientific comments:

1- The authors claim that PCPE-2 is a specific inhibitor of BMP-1/tolloid-like proteinases, based on their analysis of BMP1 and mTLL-1. This of course raises the question of the other members of the family, i.e. mTLD and TLL-2. Demonstrating restricted or pan-inhibitory properties of PCPE-2 for the whole metalloproteinase tolloid family would be a significant plus to the study.

In the new version of the manuscript, we show that mTLD is also inhibited by PCPE-2 (Fig. 6b and Supplementary Fig. 15a) and this confirms that inhibition is a common property shared by all BTPs.

Unfortunately, the quality of our mTLL-2 preparation was not good enough (see gel below) to allow proper evaluation of PCPE-2 inhibitory activity with this enzyme.

2. One of the striking findings of the present study is the potential antagonist activity of highly homologous PCPE-1 and PCPE-2, the latter being able to bind and block BMP1. I concede to the authors that clarifying the precise structural basis of PCPE-2/BMP1 interaction may fall outside the scope of the present study. However, the manuscript would gain in impact if the authors could provide further insights into the underlying mechanism. The authors show that full length PCPE-2 and PCPE-2 isolated CUB1CUB2 domains interact with BMP-1, but not full length PCPE-1. As AlphaFold models for PCPE-1 and PCPE-2 suggest distinct domain spatial distribution, I believe it would be interesting to analyze the potential interaction of BMP-1 with the isolated CUB1CUB2 domains of PCPE-1. This would provide a relevant information about a potential indirect role of the NRT domain, which may not be directly involved in the interaction, but could prevent access of BMP1 to CUB1CUB2 domains in PCPE-1.

It is true that Alpha-fold predicts a different orientation between CUB and NTR domains in PCPE-1 and -2 (see below, superimposition of the AlphaFold models of both PCPEs). However, the predicted alignment error plot shows very low confidence for the relative orientation between the CUB and NTR domains of PCPE-2. It is therefore difficult to draw conclusion regarding possible interactions between the CUB and NTR domains from these models.

AlphaFold models of PCPE-1 in tan and PCPE-2 in blue
Alignment generated using ChimeraX

Predicted alignment error plot for PCPE-2

Moreover, we have previously shown that the isolated CUB domains of PCPE-1 (up to 500 nM) do not interact with BMP-1 (Bekhouche *et al.* J Biol Chem 2010; DOI 10.1074/jbc.M109.086447). Although the hypothesis of the reviewer regarding the role of the NTR domain is interesting, we do not have any results to support this and we rather think that the role of the NTR domain in PCPE-1 (and

probably also in PCPE-2) is to bring binding partners in close proximity through NTR interaction with heparan sulfate proteoglycans (as also previously suggested in Bekhouche *et al.* J Biol Chem 2010). Finally, the revised manuscript also includes new results regarding the inhibition mechanism of BMP-1 by PCPE-2 highlighting the predominant role of CUBCUB motifs in the two proteins.

3- In this manuscript, most of the binding analyses have been achieved using SPR. Although data are broadly convincing, it should be noted that in some instance, authors chose to fit their binding data using the “Heterogeneous ligand model”, (eg in figure 4f and 4g, also in figure supp7b). I am well aware of the difficulties of fitting interaction data when using complex biomolecules in SPR, but this is less expected from a ligand such as BMP-1. I believe that this decision should be justified (in mat&meth). Was the 1:1 model was completely inappropriate, or discarded because of poor chi2 (but coherent kintetic data and Kd)?

As now clearly stated in the Materials and Methods section, our approach to choose the best model was to fit the data successively with the 1:1 binding model and the heterogenous ligand model. The latter model is possible if covalent immobilization induces different ligand presentations on the sensorchip surface which may affect interaction with the partner. In Fig. 4f and 4g, BMP-1 was immobilized at pH 5 and all its domains (having isoelectric points above 5, as depicted below) were potentially able to react with the carboxymethylated dextran matrix, leading to several possible configurations. As shown below, the 1:1 fits were not satisfactory even when some bulk effect was allowed and this is the reason why we chose to show the fit with the heterogenous ligand model.

Theoretical pI 8.75 7.85 6.07 5.19 7.83

Immobilized BMP-1 (974 RU)

Quality Control	Report	Residuals	Parameters								
Curve	ka (1/Hs)	kd (1/s)	KD (M)	Rmax (RU)	Conc (M)	tc	Flow (ul/min)	kt (RU/Hs)	RI (RU)	Chi² (RU²)	U-value
Cycle: 14 12.5 nM	3,098E+6	0,004972	1,605E-9	195,1	1,250E-8	2,004E+8	30,00	6,227E+8	0,000	96,8	4
Cycle: 16 1,5625 nM					1,563E-9		30,00	6,227E+8	0,000		
Cycle: 17 3,125 nM					3,126E-9		30,00	6,227E+8	0,000		
Cycle: 18 6,25 nM					6,250E-9		30,00	6,227E+8	0,000		
Cycle: 19 12,5 nM					1,250E-8		30,00	6,227E+8	0,000		
Cycle: 20 25 nM					2,500E-8		30,00	6,227E+8	0,000		
Cycle: 21 50 nM					5,000E-8		30,00	6,227E+8	0,000		
Cycle: 22 100 nM					1,000E-7		30,00	6,227E+8	0,000		

Quality Control	Report	Residuals	Parameters								
Curve	ka (1/Hs)	kd (1/s)	KD (M)	Rmax (RU)	Conc (M)	tc	Flow (ul/min)	kt (RU/Hs)	RI (RU)	Chi² (RU²)	U-value
Cycle: 14 12.5 nM	2,151E+6	0,002834	1,317E-9	167,5	1,250E-8	6,525E+8	30,00	2,152E+9	17,30	34,5	2
Cycle: 16 1,5625 nM					1,563E-9		30,00	2,152E+9	5,258		
Cycle: 17 3,125 nM					3,126E-9		30,00	2,152E+9	8,408		
Cycle: 18 6,25 nM					6,250E-9		30,00	2,152E+9	10,95		
Cycle: 19 12,5 nM					1,250E-8		30,00	2,152E+9	17,44		
Cycle: 20 25 nM					2,500E-8		30,00	2,152E+9	29,91		
Cycle: 21 50 nM					5,000E-8		30,00	2,152E+9	37,87		
Cycle: 22 100 nM					1,000E-7		30,00	2,152E+9	51,83		

Quality Control	Report	Residuals	Parameters								
Curve	ka (1/Hs)	kd (1/s)	KD (M)	Rmax (RU)	Conc (M)	tc	Flow (ul/min)	kt (RU/Hs)	RI (RU)	Chi² (RU²)	U-value
Cycle: 4 12.5 nM	1,595E+6	0,01459	9,150E-9	61,93	1,250E-8	2,870E+7	30,00	9,227E+7	0,000	5,11	4
Cycle: 6 1,5625 nM					1,563E-9		30,00	9,227E+7	0,000		
Cycle: 7 3,125 nM					3,126E-9		30,00	9,227E+7	0,000		
Cycle: 8 6,25 nM					6,250E-9		30,00	9,227E+7	0,000		
Cycle: 9 12,5 nM					1,250E-8		30,00	9,227E+7	0,000		
Cycle: 10 25 nM					2,500E-8		30,00	9,227E+7	0,000		
Cycle: 11 50 nM					5,000E-8		30,00	9,227E+7	0,000		
Cycle: 12 100 nM					1,000E-7		30,00	9,227E+7	0,000		

Quality Control	Report	Residuals	Parameters								
Curve	ka (1/Hs)	kd (1/s)	KD (M)	Rmax (RU)	Conc (M)	tc	Flow (ul/min)	kt (RU/Hs)	RI (RU)	Chi² (RU²)	U-value
Cycle: 4 12.5 nM	1,115E+6	0,009492	8,515E-9	53,19	1,250E-8	3,613E+7	30,00	1,123E+8	5,408	2,45	2
Cycle: 6 1,5625 nM					1,563E-9		30,00	1,123E+8	1,722		
Cycle: 7 3,125 nM					3,126E-9		30,00	1,123E+8	3,151		
Cycle: 8 6,25 nM					6,250E-9		30,00	1,123E+8	4,152		
Cycle: 9 12,5 nM					1,250E-8		30,00	1,123E+8	5,179		
Cycle: 10 25 nM					2,500E-8		30,00	1,123E+8	5,405		
Cycle: 11 50 nM					5,000E-8		30,00	1,123E+8	6,881		
Cycle: 12 100 nM					1,000E-7		30,00	1,123E+8	12,66		

In new Supplementary Fig. 8b (previously 7b) and 10a, the 1:1 binding model was used as there is no significant improvement of the fit with the more complex heterogenous ligand model.

In addition, in Supp figure 7B, the range of concentrations injected (0.78-50 nM) are below the reported K_d (80 nM using the steady state fit), which is not suitable for precise affinity determination. The risk of inaccuracy should be brought to the attention of the reader.

The reviewer is right and we have modified data presentation ($K_D = 80$ nM replaced by $K_D > 25$ nM) in new Supplementary Fig. 8b (previously 7b).

Other minor issues

1. The study shows many SPR studies using different ligand/analyte pairs, which makes it sometimes difficult to follow. The manuscript would gain in clarity if the authors could state the ligand used on the figure, for each SPR analysis.

Done

2. In Supp. Fig 5, the authors observe 2 bands for the CUB1/CUB2 domain that they claim to be 2 distinct /cub2 O-glycosylated forms. Have these O-glycosylations been described before? Did the authors confirmed this by O-Glycosidase digestion?

It was previously described by Steiglitz and colleagues (ref. 36; DOI : 10.1074/jbc.M209891200) that the conserved N-glycosylation site in the NTR domain of PCPE-2 is not used (at least when the protein is produced in HEK 293 cells) but that the protein is decorated with sialylated $\text{Gal}\beta(1-3)\text{GalNAc}$ O-linked cores, as evidenced by sialidase + endo-O-glycosidase treatment. Since these O-glycosylations have not been directly attributed to CUB1CUB2, we have moderated our statement in the legend of Supplementary Fig. 5d. Interestingly, a similar 2-band or 3-band pattern is also observed for the CUB1CUB2 domains of PCPE-1, as illustrated below.

Reviewer #3 (Remarks to the Author):

The authors present a very interesting study with potentially strong clinical significance for the field. However, in my opinion it needs to be better clarified how the difference between PCPE1 and PCPE2 in their CUB domains directs substrate specificity.

We thank the reviewer for pointing the interest and high potential impact of our study. We agree that the difference between the activities of PCPE-1 and PCPE-2 is intriguing and provide some new results to explain this difference. However, as also explained below, site-directed mutagenesis studies are presently limited by the lack of high-resolution structural data and the difficulty to make good predictions of interaction surfaces.

Major points:

1. Quite often, protease substrates, inhibitors and enhancers of protease activity create feedback loops that regulate gene expression. To that end, it would be important and interesting if the authors could investigate the mRNA expression profile of Bmp1, tll-1/2, PCPE1, betaglycan, chordin, Bmp2, Bmp4, LDLR receptor in WT and Pcolce2-null mice.

Supplementary Fig. 1 describes the mRNA expression profile of Bmp1 (2 splice variants), Tll1, Tll2 (not detected) and Pcolce. We have now also added the mRNA analysis of Tgfb1 and Bmp2, two possible targets of feedback loops, and obtained the interesting result that they are less abundant in Pcolce2-null mice. The significance of this result is discussed in the Discussion section of the revised manuscript.

2. Ref. 36 showed that PCPE2 is a procollagen C-proteinase (BMP-1 and mTLL-1) enhancer and that PCPE1 and -2 compete with procollagen C-proteinases for collagen I and II binding.

>In ref. 36 recombinant protein constructs with similar designs were compared to each other. In the present study sensitive activity tests with N- and C-terminally tagged protein constructs of PCPE1 and PCPE2 were compared to each other. Can the authors demonstrate that placement of the tags at different positions does not influence activity?

The reviewer is right to say that placement of tags can potentially affect activity. We are now providing evidence that the presence/position/length of tag has no substantial effect on the activity of PCPE-1 and -2 (Supplementary Fig. 6b and c). The fact that the untagged CUB domains of PCPE-2 behave very similarly to tagged PCPE-2 is also an argument in favor of the absence of effect of the His tag.

>also often times not the appropriate controls were used: for instance in several Figs. CUB or NTR domains of PCPE2 were compared to PCPE1 full length and not to CUB and NTR domains of PCPE1.

Can the authors reproduce the same results with the appropriate controls next to each other? It is possible that expression of separate domains lead to conformational changes which lead to different functional properties.

Can the authors provide CD measurements to assure that the conformation of CUB and NTR domains are preserved?

We draw the attention of the reviewer to the fact that CUB and NTR domains were not expressed independently but as a full-length protein (PCPE-2 3C) which was subsequently cleaved to generate the independent CUB and NTR domains. CD measurements were performed before and after cleavage of PCPE-2 3C and the spectra were perfectly conserved (new Supplementary Fig. 6a).

We now also show the effect of PCPE-1 CUB and NTR domains next to PCPE-2 domains in Fig. 4a. It was unfortunately not possible to do the same thing for all figures as some of them are already very busy. We kindly refer the reviewer to previously published work (Kronenberg *et al.* J. Biol. Chem. 2009 : DOI 10.1074/jbc.M109.046128).

3. The most interesting aspect of the paper is that apparently CUB domains of PCPE1 and PCPE2 are responsible for their different behavior, although their conformation seems similar (Fig2D). Can the authors pinpoint the exact structural features present in the CUB domains of PCPE2 that lead to inhibition instead of enhancement of processing as already shown for the CUB domains of PCPE1? In

Fig. 2D a superposition is shown. However, an alphafold model or CD measurements of CUB domains from both PCPE1 and PCPE2 together with the sequence alignments shown in suppl. Fig6 would allow the authors to identify the crucial residues responsible for the observed differences in functional behavior. If indeed a stronger affinity of the PCPE-2 CUB domains to BMP-1 is the reason for the BMP-1 inhibition of processing, then site-directed mutagenesis should be employed to neutralize the interaction of PCPE-2 to BMP-1. The generated PCPE-2 mutant should be assessed for inhibition or enhancement of BMP-1 proteolytic processing. CD spectra measurements should be carried out to ensure that the introduced mutations do not affect PCPE2 folding but specifically loosen its interaction with BMP-1. The authors have gathered extensive experience about the structure-function relationship of CUB domains of PCPE1 as evidenced by ref.2 of the supplementary references and have the knowledge and expertise to carry out these analyses.

In the revised version of the manuscript, we provide several new important elements regarding the mechanism of action of PCPE-2. In particular, through domain deletion experiments, we were able to identify the domains driving the formation of the inhibitor/enzyme complex for both BMP-1 and PCPE-2 (new Fig. 4-5 and Supplementary Fig. 10-14). These results show that both CUB domains in PCPE-2 are necessary while, in BMP-1, the interaction also involves the first two CUB domain. We also provide more compelling evidence that PCPE-2 does not bind in the active site directly (through competition with a small-molecule inhibitor; Supplementary Fig. 12c) but our results do not rule out the possibility of an interaction with another part of the catalytic domain.

To go further and try to pinpoint the residues involved in PCPE-2/BMP-1 interaction, we have first generated a model of the BMP-1[cat-CUB1CUB2]/PCPE-2[CUB1CUB2] complex with AlphaFold as shown below. However, the confidence score of this model is low (0.43) and it does not fit with our experimental data as the CUB2 domain of BMP-1 is far away from the interaction site while we know that it is required for inhibition (Fig. 4c). Therefore, this model cannot be used to select appropriate residues for mutagenesis studies.

Nevertheless, as suggested by the reviewer, we also conducted the same type of sequence analyses for the CUB domains of PCPE-2 as those described in 2007 for the CUB domains of PCPE-1 (Blanc *et al.* DOI : 10.1074/jbc.M701610200). We found 4 charged residues located in flexible loops that were conserved in mammalian PCPE-2 proteins but were absent in all other analyzed CUB domains (including those from PCPE-1) suggesting that they could be specifically involved in BMP-1 binding. These residues were mutated to alanine and one triple mutant (K46A/E197A/R220A) led to

substantially reduced binding and inhibitory activity of the CUB domains of PCPE-2 (Fig. 5c-f; Supplementary Fig. 14).

It was however not possible to perform CD measurements for all the new mutants (deletions + point mutations) because we only made small-scale productions of each of them. Also, PCPE-2-derived proteins are stored in an optimized storage buffer that is not compatible with CD due to high salt concentration (0.5 M NaCl) and they usually do not behave very well upon dialysis.

4. Conceptionally, it should be made better clear to the audience why PCPE2 acts as an inhibitor. The only logical explanation is that it interacts either with different affinities to different substrates or as the authors pointed out it is that the kinetics of the tripartite complex formation of the substrates and BTPs are significantly altered. In my opinion more work needs to be invested to better answer this question. Affinities of CUB domains of PCPE1 and PCPE2 with different substrates should be determined. Also tripartite complex formation should be exemplarily shown for some critical substrates, such as procollagen II versus procollagen III.

We have also checked the competition with mini-procollagens I and II and found that they can displace PCPE-2 from BMP-1 (Supplementary Fig. 13a). However, we do not think that the interaction of PCPE-2 with BMP-1 substrates is a general mechanism to explain its inhibitory activity. It occurs with procollagens and we have seen that it leads to this unusual dual activity which is not observed with other BMP-1 substrates. To further ascertain this hypothesis, we have also tested the interaction of PCPE-2 with betaglycan and with the ectodomain of LDLR (Supplementary Fig. 13b; no interaction).

Minor: In the working model also the specific functions of the different functional domains should be illustrated

We have modified the model to integrate the new data on the interaction of PCPE-2 with BMP-1 (new Fig. 7).

REVIEWERS' COMMENTS

Reviewer #1 (Remarks to the Author):

The authors have addressed some of the key issues that have been raised and have made modest inroads into the potential biological relevance of their findings. Both the original and the new biochemical studies have been carefully executed in an undeniably challenging system. However, demonstrating the roles of this potentially important inhibitor PCPE-2 in determining tolloid-like proteinases, growth factor etc. regulation in vivo, in both biological and clinical settings, will be essential to justify further interest.

Reviewer #2 (Remarks to the Author):

I have reviewed the revised version of the manuscript from Vadon-Le Goff et al. (ref NCOMMS-22-34298), entitled "Identification of PCPE-2 as the first endogenous specific inhibitor of human BMP-1/tolloid-like proteinases".

The authors have addressed my major concerns and I find this new version of the article much improved. Given the quality of the experimental work and the originality and impact of the data, I have now no reservations regarding publication of the present manuscript in Nature communications.

Reviewer #3 (Remarks to the Author):

All my concerns were well addressed by the authors. Therefore in my opinion the manuscript can be accepted for publication.